# TSG-6+ cancer-associated fibroblasts modulate myeloid cell responses and impair anti-tumor response to immune checkpoint therapy in pancreatic cancer

Swetha Anandhan [1,2,3], Shelley Herbrich[1,3], Sangeeta Goswami[1,3,4],
Baoxiang Guan[1,3], Yulong Chen[5], Marc Daniel Macaluso[5], Sonali Jindal [5],
Seanu Meena Natarajan[4], Samuel W. Andrewes[1,2,3], Liangwen Xiong[1,3],
Ashwat Nagarajan[1,3], Sreyashi Basu [5], Derek Ng Tang[1,3], Jielin Liu[1,2,3],
Jimin Min[6,7], Anirban Maitra [6,7] & Padmanee Sharma [1,3,4,5] ✉

Resistance to immune checkpoint therapy (ICT) presents a growing clinical challenge. The tumor microenvironment (TME) and its components, namely tumor-associated macrophages (TAMs) and cancer-associated fibroblasts (CAFs), play a pivotal role in ICT resistance; however, the underlying mechanisms remain under investigation. In this study, we identify expression of TNF-Stimulated Factor 6 (TSG-6) in ICT-resistant pancreatic tumors, compared to ICT-sensitive melanoma tumors, both in mouse and human. TSG-6 is expressed by CAFs within the TME, where suppressive macrophages expressing *Arg1*, *Mafb*, and *Mrc1*, along with TSG-6 ligand *Cd44*, predominate. Furthermore, TSG-6 expressing CAFs co-localize with the CD44 expressing macrophages in the TME. TSG-6 inhibition in combination with ICT improves therapy response and survival in pancreatic tumor-bearing mice by reducing macrophages expressing immunosuppressive phenotypes and increasing CD8 T cells. Overall, our findings propose TSG-6 as a therapeutic target to enhance ICT response in non-responsive tumors.

Immune checkpoint therapy (ICT) has revolutionized the clinical outcomes of cancer patients, leading to long-term durable responses in cancers such as melanoma[1]. However, certain tumor types, such as pancreatic ductal carcinoma (PDAC), have shown poor response to this treatment[2,3]. The mechanisms underlying why specific tumor types respond to ICT while others do not are still not well understood. Understanding these resistance mechanisms is crucial to developing rational therapeutic combinations with ICT that can help convert "cold tumors" to "hot tumors"[4].

The tumor microenvironment (TME) plays an undisputed role in determining responses to ICT. A high abundance of intratumoral myeloid cells, such as tumor-associated macrophages, have been correlated with unfavorable outcomes and reduced therapeutic efficacy[5–8]. Corroborating this, our group previously compared

[1]Department of Genitourinary Medical Oncology, The University of Texas MD Anderson Cancer Center, Houston, TX, USA. [2]The University of Texas MD Anderson Cancer Center UTHealth Houston Graduate School of Biomedical Sciences, The University of Texas MD Anderson Cancer Center, Houston, TX, USA. [3]The James P. Allison Institute, The University of Texas MD Anderson Cancer Center, Houston, TX, USA. [4]Department of Immunology, The University of Texas MD Anderson Cancer Center, Houston, TX, USA. [5]Immunotherapy Platform, The University of Texas MD Anderson Cancer Center, Houston, TX, USA. [6]Department of Translational Molecular Pathology, The University of Texas MD Anderson Cancer Center, Houston, TX, USA. [7]Sheikh Ahmed Center for Pancreatic Cancer Research, The University of Texas MD Anderson Cancer Center, Houston, TX, USA. ✉e-mail: PadSharma@mdanderson.org

immune infiltrates between ICT-sensitive melanoma and ICT-resistant pancreatic patient tumors and found a significant abundance of suppressive myeloid cells in the pancreatic stroma, which were absent in melanoma[9]. This association between myeloid cells and response to ICT has been primarily attributed to the capacity of these cells to suppress T cell function and facilitate tumor growth in response to environmental cues[10–12]. For this reason, multiple strategies to directly target myeloid recruitment, polarization, and function are currently being evaluated clinically and preclinically[13]. However, such approaches have achieved limited clinical success due to myeloid plasticity, tumor-specific heterogeneity, and the lack of validated clinical measures of myeloid functionality[14–17]. Therefore, alternative strategies to target myeloid cell function are necessary to improve therapy efficacy.

Cancer-associated fibroblasts (CAFs) constitute the primary component of the stromal compartment and have been implicated in promoting tumor growth through various mechanisms, including the regulation of suppressive functions and recruitment of intratumoral myeloid cells via TGFβ, IL-6, GM-CSF, reactive oxygen species (ROS) and other chemokines[18–21]. Although CAFs appear to be promising candidates for enhancing therapy responses, depleting them may not effectively stimulate anti-tumor responses. This was substantiated by studies demonstrating that CAF depletion paradoxically facilitated the acceleration of pancreatic tumor growth by promoting epithelial-mesenchymal transition[22,23]. Furthermore, the application of single-cell RNA sequencing (scRNAseq) has unveiled distinct CAF subtypes with diverse functions in the TME, emphasizing the complex nature of CAF-mediated regulation of immune cell function[24–26]. Consequently, targeting the CAF-specific regulation of immune cell function may represent a more efficacious approach.

In this work, we perform scRNAseq on both the immune and non-immune compartments of the TME, revealing a CAF-secreted myeloid cell regulating mediator, called TNF-stimulated gene 6 (TSG-6) enriched in pancreatic tumors compared to melanoma tumors, in both murine models and patient samples. Furthermore, pancreatic tumors are enriched in suppressive macrophages expressing the known TSG-6 receptor, *Cd44*, which co-localize with TSG-6 within these tumors. Importantly, in vivo neutralization of TSG-6 in combination with immune checkpoint antibodies decreases suppressive macrophage subsets and increases CD8 T cells in the tumor, correlating with improved survival. Overall, using a reverse translational approach, we have identified a mechanism of ICT resistance via the CAF-secreted mediator TSG-6. We propose that TSG-6 is a rational therapeutic target that can be used in combination with ICT to improve clinical responses in fibrotic tumor types.

## Results

### TSG-6 expression is higher in pancreatic tumors versus melanoma

To perform a comparative analysis, we employed B16F10 (melanoma) and mT4 (PDAC) as ICT-sensitive and ICT-resistant tumor models, respectively (Supplementary Fig. 1a, b). To investigate the role of the tumor and stromal compartment in regulating ICT resistance, we first conducted scRNAseq on CD45-negative cells sorted from orthotopic B16F10 and mT4 tumors (Fig. 1a, Supplementary Fig. 1c, Supplementary Data 1). Cell clusters obtained were characterized based on their gene expression profile and identified as B16F10 tumor cells (*Pmel*, *Mlana*), mT4 tumor cells (*Krt18*, *Krt19*), and fibroblasts (*Col1a1*, *Dcn*) (Fig. 1b, c). Of note, we observed higher abundance of fibroblasts in mT4 tumors compared to B16F10 tumors which is consistent with observations from patient tumors (Fig. 1d, Supplementary Fig. 1d–f). Analysis of the gene expression profile in the CAFs revealed elevated expression of the gene *Tnfaip6* (encoding the protein TSG-6) within the mT4 tumors (Fig. 1e). Further, analysis of the TCGA datasets of melanoma and pancreatic cancer patients indicated significantly

higher expression of *TNFAIP6* gene in pancreatic tumors compared to melanoma tumors highlighting the relevance of our finding in human setting (Fig. 1f). Together, our data highlights that CAF-secreted TSG-6 is elevated in pancreatic tumors compared to melanoma.

### TSG-6 expression is induced in cancer setting

Next, we wanted to determine whether TSG-6 was expressed constitutively or induced during tumor development. For this, we analyzed a publicly available human scRNAseq dataset comprising tissues from 24 pancreatic tumors and 11 normal pancreas[27] (Fig. 2a–e, Supplementary Fig. 2a, b). Consistent with our murine findings, we observed that *TNFAIP6* expression was enriched predominantly in CAFs, which were absent in the normal pancreas (Fig. 2d, e). Importantly, this data indicated that TSG-6 expression is induced only in the tumor setting. To determine if TSG-6 protein levels in the TME correlate with the gene expression data, we performed multi-immunofluorescence (mIF) on tissue samples from patients with pancreatic cancer and melanoma. Corroborating with our above findings, we observed higher abundance of TSG-6 expressing CAFs in the pancreatic TME when compared to melanoma (Fig. 2f–h, Supplementary Table 1). Overall, our data highlights that TSG-6 is induced in cancer and enriched in the ICT-resistant pancreatic tumors as compared to ICT-sensitive melanoma, both in murine models and patient samples.

### Pancreatic tumors are dominated by suppressive myeloid cells

To investigate whether TSG-6 expressed in the TME regulates immune cell polarization and function in tumors, we characterized the intratumoral immune cell landscape in the B16F10 and mT4 tumors using scRNAseq (Fig. 3a). The analyzed CD45+ cell populations encompassed diverse immune subsets, including T cells, NK cells, B cells, as well as myeloid cells such as macrophages, dendritic cells, and neutrophils (Fig. 3b–d, Supplementary Fig. 3a, b).

We found that the total frequency of macrophages, monocytes and neutrophils was nearly two-fold higher in mT4 tumors compared to B16F10 tumors (Fig. 3c). These differences in cell frequencies were further validated at the protein level (Supplementary Fig. 3c). Importantly, similar to our human findings, none of the immune cells expressed *Tnfaip6* (Supplementary Fig. 3d). Of all the immune subsets present in both the tumor models, the macrophage subsets in mT4 tumors were transcriptionally distinct from those in B16F10 tumors, as evidenced by the presence of discrete clusters between the two tumor models (Fig. 3c) and their non-overlapping distribution on the UMAP plots (Fig. 3e, f). Given that macrophage heterogeneity has been linked to ICT resistance, we further investigated the phenotypic differences between the macrophage subsets. We identified two macrophage clusters that were prevalent in mT4 tumors: one which expressed *Mmp14*, *Axl*, and *Mafb* (Mafb+ macs), and the other which expressed *Vegfa*, *Arg1*, *Ccl24*, and *Fn1* (Arg1+ macs) (Fig. 3g). Expression of these genes in macrophages have been primarily associated with immune suppressive phenotypes[28,29]. In contrast, the dominant macrophage cluster in B16F10 tumors (Cd72+ macs) expressed the antigen-presenting gene (*Cd72*), immune cell migratory chemokines (*Ccl2*, *Ccl7*), and interferon-induced genes (*Cxcl10*, *Isg15*). Only a small macrophage cluster expressed tumor-promoting genes (*Csf1r*, *Selenop*) (Csf1r+ macs). This data indicates that most macrophages in B16F10 tumors have a pro-inflammatory phenotype. GSEA showed that mT4 macrophages were enriched in TGFβ and TNFα signaling via NF-kB pathways, while B16F10 macrophages were enriched in interferon-gamma and alpha response pathways, indicating their opposing functions in the tumor (Fig. 3h). These observations suggested that the pancreatic TME, which has elevated TSG-6, is also dominated by suppressive macrophages in contrast to the B16F10 melanoma TME.

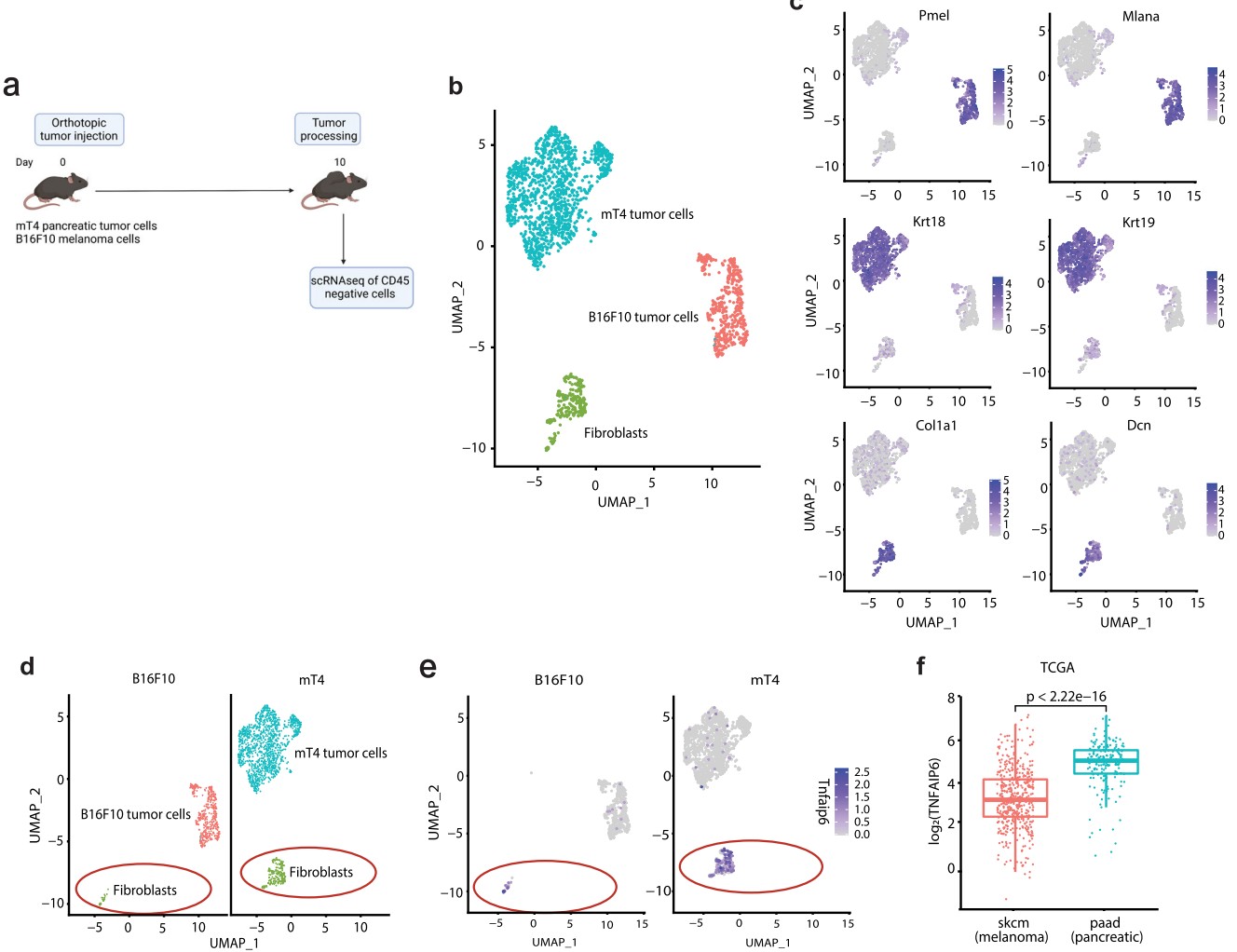

**Fig. 1 | TSG-6 gene expression is higher within pancreatic tumors when compared to melanoma tumors. a** Schematic representation of the scRNAseq experimental design created with BioRender.com, released under a Creative Commons Attribution-NonCommercial-NoDerivs 4.0 International license. 3 tumors in each group were pooled for internal control. **b** Representative Uniform Manifold Approximation Projection (UMAP) plot of sorted intratumoral CD45-negative cells. Each dot represents a cell. **c** UMAP plots indicating expression of genes depicting B16F10 tumor cells (*Pmel*, *Mlana*), mT4 tumor cells (*Krt18*, *Krt19*) and fibroblasts (*Col1a1*, *Dcn*). **d** UMAP plots highlighting differences in fibroblast abundance between mT4 and B16F10 tumors (red circle). **e** UMAP plot depicting

*tnfaip6* (gene encoding TNF Stimulating Gene-6 (TSG-6)) expression in B16F10 and mT4 tumors (red circle). **f** Box-and-whisker plot representing *TNFAIP6* RNA expression in TCGA datasets of melanoma (skcm, skin cutaneous melanoma) (*n* = 480 patients) and pancreatic patient tumors (paad, pancreatic adenocarcinoma) (*n* = 186 patients). Each dot represents a patient. Statistical significance was calculated using Student's *t* test (two-tailed) and *p* value for the comparison has been indicated in the figure. Data are presented as mean values ± SD. The center of the plot represents mean of the group and the whiskers represent minimum-maximum values.

## TSG-6+ CAFs co-localize with CD44+ macrophages in pancreatic tumors

To assess whether TSG-6 plays a role in the distinct macrophage phenotypes seen in the two tumor models, we examined the expression of *Cd44*, the only known ligand of TSG-6, across all immune cells. Our observations indicated an enrichment of *Cd44* in mT4 macrophages compared to B16F10 macrophages, suggesting that macrophages may serve as important interacting partners of TSG-6 in the pancreatic TME (Fig. 4a, b, Supplementary Fig. 4a, b). Macrophages within patient tumors also exhibited elevated levels of *CD44* compared to those present in normal pancreas, corroborating our murine findings (Supplementary Fig. 4c). To determine if these correlations translated to in vivo interactions, we performed multi-immunofluorescence (mIF) on baseline pancreatic patient tissues (Fig. 4c–e). Our analysis revealed the presence of TSG-6 surrounding CD68+ myeloid cells, as well as the co-expression of TSG-6 with CD44+ CD163+ CD68+ cells within the tumors (white

arrow, Fig. 4c). To quantify these observations, we conducted infiltration analysis on the images and observed that TSG-6 expressing CAFs were in closer proximity to CD44+ myeloid cells compared to CD44- myeloid cells (Fig. 4d, e). Together, our data above suggests that TSG-6 co-localize with macrophages within the tumors.

## Inhibition of TSG-6 improves ICT efficacy in mice

Since TSG-6 has been shown to regulate macrophage recruitment and M2-like polarization in other inflammatory settings[30,31], we aimed to investigate whether inhibiting TSG-6 function could reverse myeloid suppression and create a favorable TME for improved ICT response. To evaluate the translational potential of TSG-6 inhibition, we employed a commercially available anti-TSG-6 antibody and conducted survival experiments in the pancreatic tumor model (Fig. 5a). Due to the highly aggressive nature of the mT4 tumor model, we utilized the previously established single clone mT4-LS tumor cells derived from the mT4

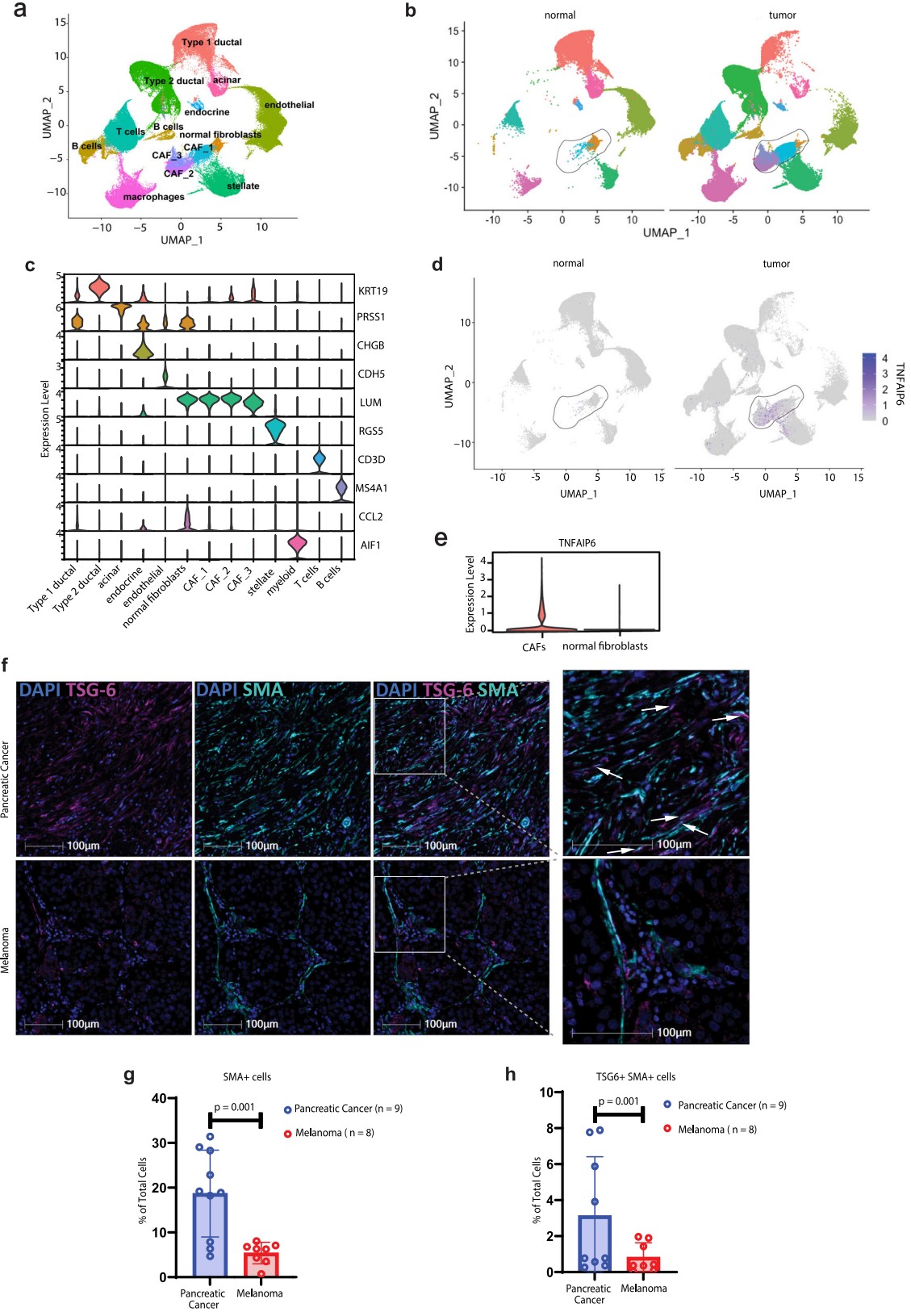

parental line, which offer a broader therapeutic window[32]. Mice treated with combination of anti-CTLA-4 and anti-PD-1 (combo ICT) and anti-TSG-6 antibodies had significantly longer survival than those treated with ICT or anti-TSG-6 alone (Fig. 5b). To investigate the changes induced by this combinatorial treatment on the intratumoral immune cells, we performed mass cytometry (CyTOF). We observed a decrease in suppressive macrophages (VISTA + CD206+) and regulatory T cells (Tregs), along with a concurrent increase in the abundance of CD8 T cells in the tumors treated with anti-TSG-6 and ICT antibodies (Fig. 5c–e). Overall, the above data suggests that neutralizing TSG-6 reinvigorates the TME, leading to improved ICT efficacy in murine pancreatic cancer.

**Fig. 2 | TSG-6 expression is induced in cancer setting. a** UMAP plot of tumor and stromal compartment from 24 PDAC patients and 11 normal pancreas reanalyzed from ref. 27. **b** UMAP plots indicating distribution of all cells across the two groups (normal pancreas and tumor). Enclosed region in black depicts the fibroblasts present in normal pancreas and tumors. **c** Violin plot indicating the markers that were used to define the cell subsets in (**a**). **d** Expression of *TNFAIP6* across all cells present in normal pancreas and tumors. Enclosed region in black depicts the fibroblasts present in normal pancreas and tumors. **e** Violin plot depicting the quantification of *TNFAIP6* expression in cancer-associated fibroblasts (CAFs) and normal fibroblasts. **f** Representative multi-immunofluorescence (mIF) images

demonstrating presence of TSG-6 protein in human pancreatic and melanoma FFPE samples. Zoomed vision of the images are shown on the right and white arrows highlight the TSG-6+ SMA+ cells in the pancreatic TME which are absent in the melanoma tumors. **g** Bar plot representing quantification of SMA+ cells and (**h**) TSG-6+ SMA+ cells in pancreatic (*n* = 9 patients) and melanoma (*n* = 8 patients) tissues. Data are presented as mean values ± SD. Statistical significance was calculated using Student's *t* test (two-tailed) and *p* values for each comparison has been indicated in the figure. The center of the plot represents mean of the group and the whiskers represent minimum-maximum values. Source data are provided as a Source Data file.

## Discussion

In this study, we have identified TSG-6, a protein expressed by pancreatic CAFs that promotes ICT resistance through macrophage-mediated immunosuppression. Our findings indicate that inhibiting TSG-6 function can reprogram the TME, leading to an improved response to ICT in mice (Fig. 6).

TSG-6 is a 30-kDa secreted protein classified within the hyaluronic acid-binding protein family and plays a role in inflammation regulation and maintenance of extracellular matrix production during homeostasis[33,34]. In chronic inflammatory settings, TSG-6 interacts with CD44, its ligand on myeloid cells, and regulates macrophage recruitment and M2-like polarization[30,31]. TSG-6 is synthesized in response to proinflammatory mediators such as TNFα and TGFβ[35], pathways which we show are exclusively upregulated by the suppressive myeloid cells in the pancreatic TME. Our data suggest that TSG-6 secretion could create a feedforward loop where CAFs secrete TSG-6, which polarizes myeloid cells to express TNFα and TGFβ, thereby inducing more TSG-6 secretion. This process may progressively polarize intratumoral myeloid cells towards an immunosuppressive phenotype, promoting tumor progression. Here, it is important to note that while CD44 is expressed by multiple cell types, in this study we focus on the role of TSG-6 induced immunosuppression via macrophages. Our data also indicates high *Cd44* expression in neutrophils present in pancreatic tumors (Supplementary Fig. 4a). Thus, the mechanism of TSG-6-mediated myeloid suppression and other CD44 expressing non-immune cells in cancer warrants further investigation.

The therapeutic effect of TSG-6 has been well documented in various inflammatory diseases such as arthritis, peritonitis, corneal injury and colitis[30,36–38]. However, role of TSG-6 in cancer is not well understood. To our knowledge, only two studies have demonstrated the role of TSG-6 in tumor promotion and metastasis[39,40]. In this study, we show that CAF-expressed TSG-6 plays a crucial role in ICT resistance. Notably, we observe that all CAFs in the tumor express TSG-6, regardless of the subset type present, making it an ideal candidate for targeting. It is important to highlight that significantly higher abundance of CAFs are present in pancreatic tumors, which correlates to increased expression of TSG-6 in these tumors as compared to melanoma. Furthermore, the predominant CAF subsets in these pancreatic tumors appear to be myofibrotic (Supplementary Fig. 5), consistent with observations from a previous study[41]. Given that smooth muscle actin (SMA) is abundantly expressed in myofibrotic CAFs (myCAFs), we specifically evaluated TSG-6 in SMA+CAFs, recognizing varying expression levels in other CAF subsets[41]. Additionally, while SMA can also be expressed by other cell subsets such as stellate cells, our data suggests that CAFs are the predominant source of TSG-6 and therefore we posit that SMA + TSG-6 expressing cells are CAFs. Lastly, TCGA screening showed that expression levels of TSG-6 observed in pancreatic tumors is similar in other ICT non-responsive tumor types, such as glioblastoma and sarcoma (Supplementary Fig. 6). A recent study suggests that CAFs are conserved across tumor types[42], implying that TSG-6 could have a similar mechanism of ICT resistance in these tumor types, which requires further exploration.

Multiple studies, including those from our group, have indicated the existence of myeloid heterogeneity in the TME and showed that

specific myeloid subsets present in the TME carry out suppressive functions in a tumor-specific manner[13,14,43]. Our study identified distinct myeloid cells in tumors that respond versus those that do not respond, indicating the importance of assessing baseline immune population to evaluate treatment efficacy. Additionally, our earlier study highlighted presence of abundant VISTA+ myeloid cells in the pancreatic stroma compared to melanoma[9]. Here, we observed that inhibition of TSG-6 in combination with ICT decreased abundance of VISTA+ CD206+ suppressive myeloid cells in tumor. Therefore, TSG-6 could be an ideal combinatorial candidate with checkpoint antibodies to improve therapy responses.

Overall, analysis of human data with correlative single-cell analyses of the immune and stromal compartment between an ICT-responsive melanoma and non-responsive pancreatic murine model has identified a TSG-6-mediated myeloid suppressive pathway that induces ICT resistance. Our data provide a strong rationale to target TSG-6 to improve ICT responses in TSG-6 expressing and myeloid-rich tumor types.

## Methods

### Ethics statement
This research complies with all relevant ethical regulations. The clinical protocol was approved by the internal review board at The University of Texas MD Anderson Cancer Center. All animal experiments were conducted according to protocols approved by the Animal Resource Center at The University of Texas MD Anderson Cancer Center.

### Patients and surgical samples
Patient samples were collected after appropriate informed consent obtained on MD Anderson internal review board-approved protocol no. PA13-0291. No financial compensation was provided for participation in the trial protocol. All patients signed informed consent for participation in PA13-0291 before surgery or sample collection. The clinical characteristics of individual patients are shown in Supplementary Table 1.

### Mice
C57BL/6 (5–7 weeks) mice were purchased from the National Cancer Institute (Frederick, MD). Age- and sex-matched mice were used for each experiment. Since the study involves the understanding of CAF mediators in ICT responses, sex- and gender-based analyses were not performed. All mice were kept in specific pathogen-free conditions in the Animal Resource Center at The University of Texas MD Anderson Cancer Center. The mice were maintained at 20–26 °C, 30–70% humidity and under a 12/12 h light/dark cycle. Animal protocols were approved by the Institutional Animal Care and Use Committee (IACUC) of The University of Texas MD Anderson Cancer Center.

### Cell lines and tumor models
mT4 pancreatic cell line was a generous gift from Dr. David A. Tuveson (Cold Spring Harbor Laboratory, NY). mT4 is an organoid cell line generated from mouse pancreata containing PDAC from the *Kras*^LSL-G12D/+^;*Trp53*^LSL-R172H/+^;*Pdx1–Cre* mouse model under C57BL/6

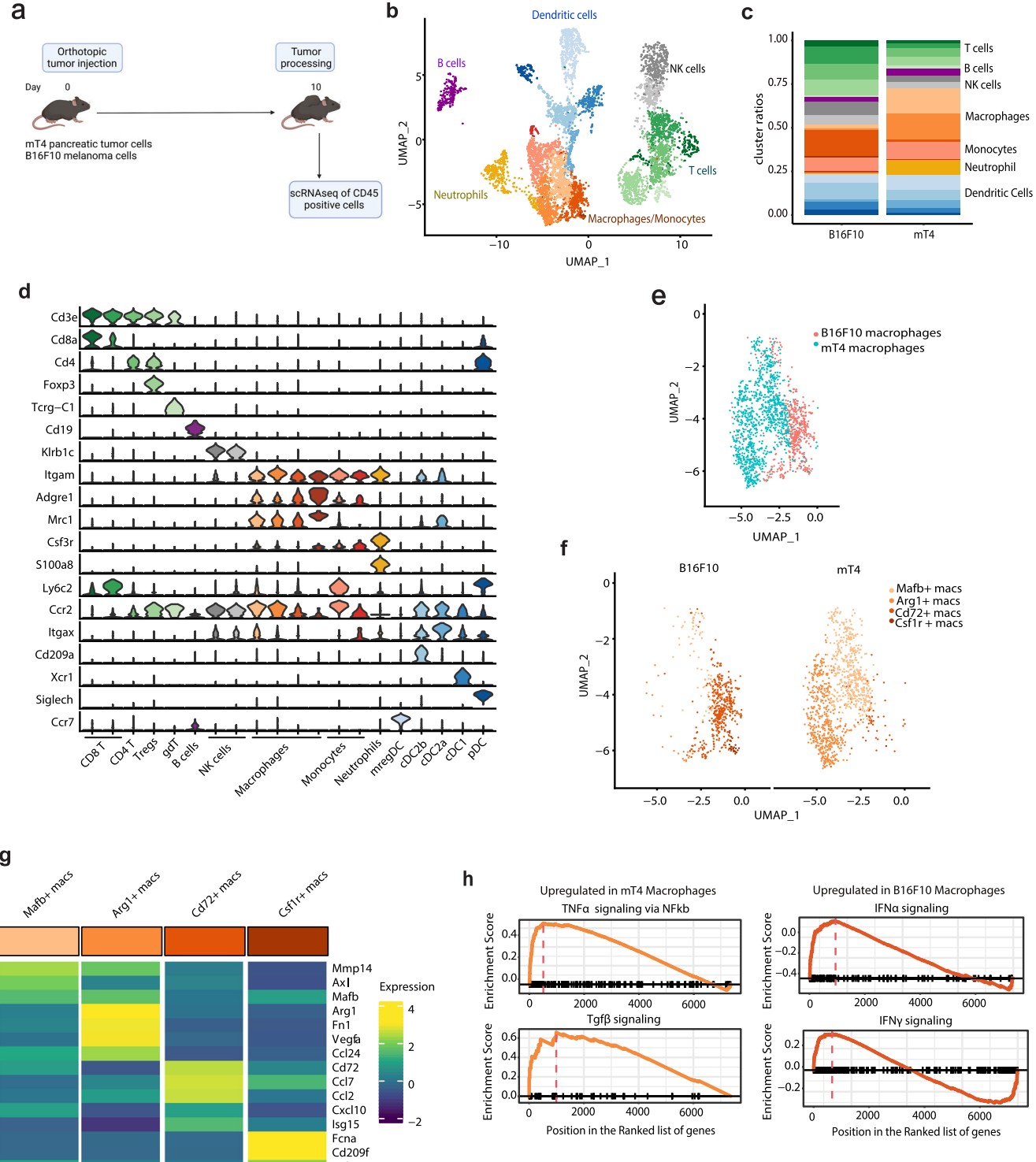

**Fig. 3 | Pancreatic tumors are dominated by suppressive myeloid cells.**
**a** Schematic representation of the scRNAseq experimental design created with BioRender.com, released under a Creative Commons Attribution-NonCommercial-NoDerivs 4.0 International license. **b** Representative landscape in B16F10 and mT4 tumors. Three tumors in each group were pooled for internal control. All major immune cell subsets were identified. **c** Cluster frequency plot of each immune subset in B16F10 and mT4 tumors. The T cells are depicted in shades of green, B cells in purple, NK cells in gray, macrophages and monocytes in red, neutrophil in orange, and dendritic cells in blue. **d** Violin plot representing expression of marker genes used for characterization of immune subsets identified in (**b**). **e** UMAP plots depicting total macrophages in B16F10 tumors (red) and mT4 (blue) to highlight minimal overlap between subsets. Each dot represents a cell. **f** Distribution of the macrophages across the B16F10 and mT4 tumors depicted in (**e**). **g** Heatmap of functional markers for the individual macrophage subsets providing phenotypic information. Expression levels are scaled between minimum and maximum expression for each gene across all clusters. **h** GSEA results depicting differential pathways between mT4 and B16F10 macrophages.

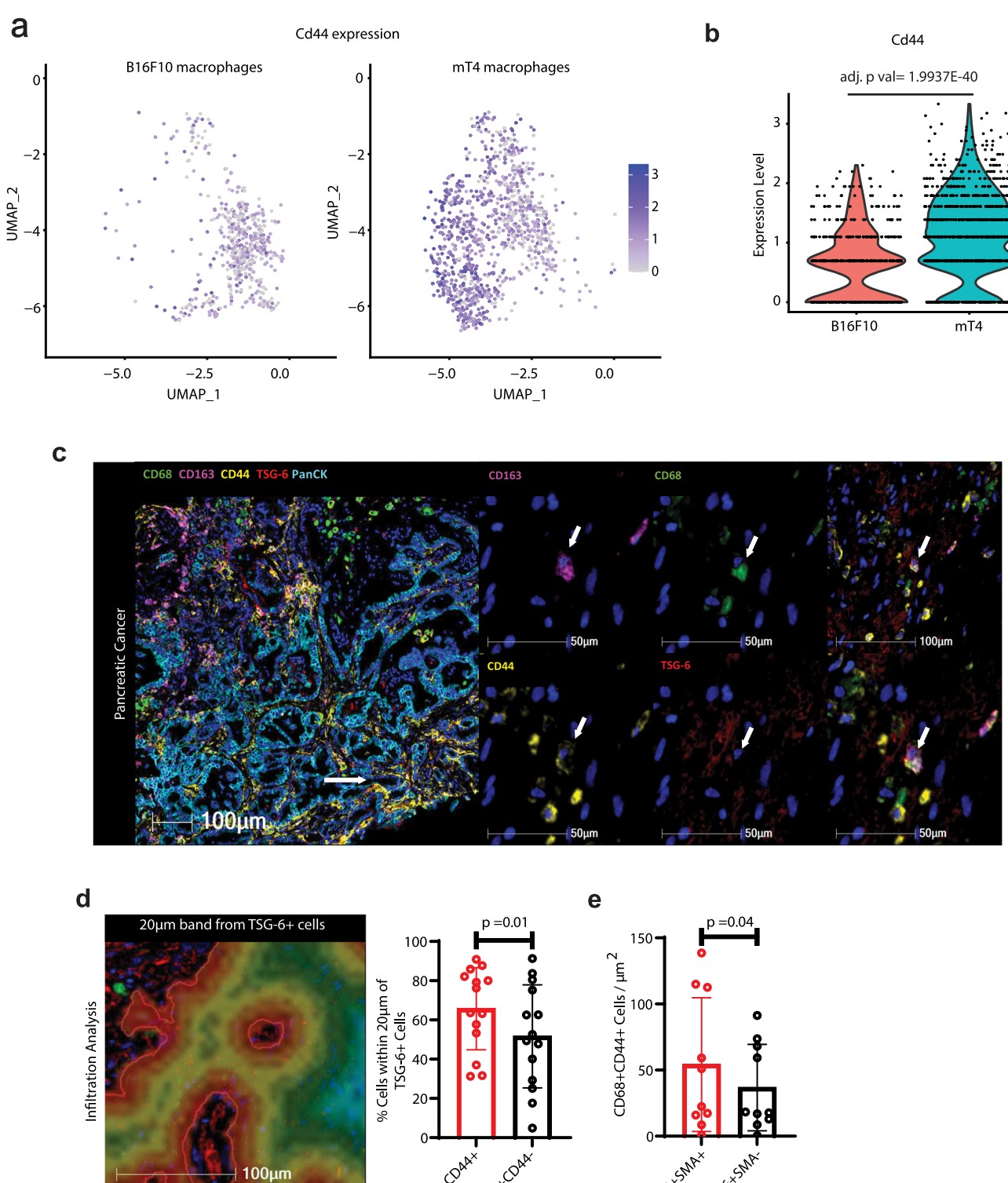

**Fig. 4 | TSG-6 expressing cancer-associated fibroblasts co-localize with CD44+ macrophages in pancreatic tumors. a** Expression of *Cd44* across macrophages present in B16F10 and mT4 tumors. **b** Violin plot quantifying *Cd44* expression in macrophages present in B16F10 and mT4 tumors. **c** Representative multi-immunofluorescence (mIF) image highlighting co-localization of CD68+ CD44+ CD163+ myeloid cells with TSG-6 (white arrow) in human pancreatic tissue FFPE samples. **d** Quantification of the mIF images using infiltration analysis technique. Red borders indicate TSG-6+ cells and areas from red to green indicate the increasing distance from the TSG-6+ cells (green being furthest). Percentage of

CD68+ CD44+ cells that were at a distance of 0–20 µm (closest) from TSG6+ cells were quantified, and bar plotted (*n* = 14 pancreatic tissues; patient characteristics provided in Supplementary Table 1). **e** Quantification of number of CD68+ CD44+ cells that were at a distance of 0–20 µm (closest) from TSG-6+ SMA+ versus TSG-6+ SMA- cells (*n* = 10 pancreatic tissues; patient characteristics provided in Supplementary Table 1). Each symbol represents a patient. Statistical significance was calculated using Student's *t* test (two-tailed). Data are presented as mean values ± SD and *p* values for each comparison has been indicated in the figure. Source data are provided as a Source Data file.

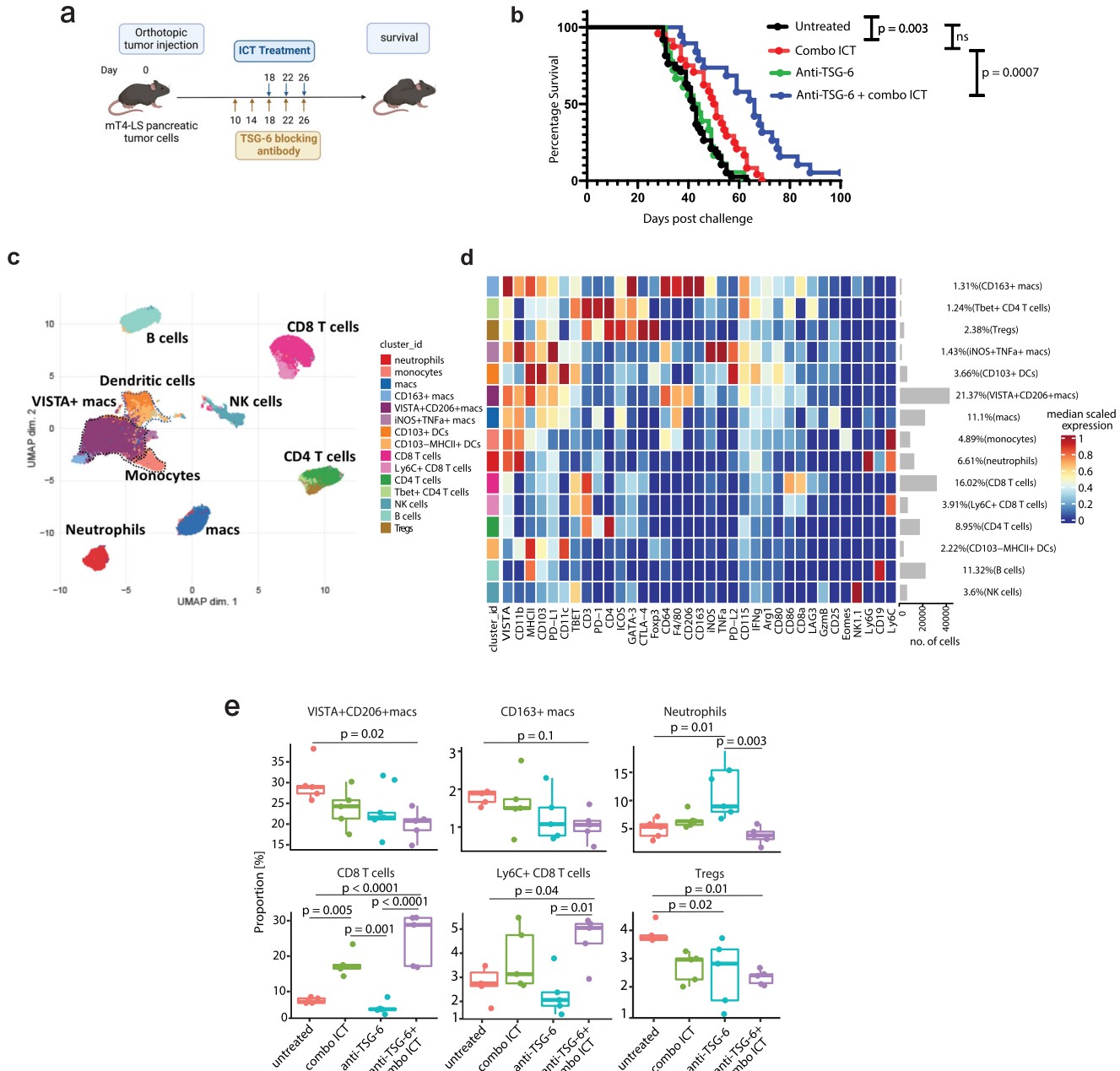

**Fig. 5 | Inhibition of TSG-6 improves ICT efficacy in mice. a** Representation of experimental design for the in vivo antibody blocking studies performed, created with BioRender.com, released under a Creative Commons Attribution-NonCommercial-NoDerivs 4.0 International license. **b** Kaplan-Meier survival plot indicating therapeutic activity of anti-TSG-6, anti-CTLA-4 and anti-PD-1 in pancreatic tumor-bearing mice. Data cumulative of three independent experiments ($n = 43$ mice in untreated group, $n = 23$ mice in combo ICT treated group, $n = 21$ mice in anti-TSG-6 treated group, $n = 19$ mice in anti-TSG-6+ combo ICT treated group). Statistical significance was calculated using Log-rank Mantel-Cox test (two-sided). **c** UMAP representation of intratumoral immune cells identified upon CyTOF analysis. **d** Heatmap indicating expression of proteins analyzed in the CyTOF experiment and phenotypic characterization of each cluster identified and represented in (**c**). Expression levels are scaled between minimum and maximum expression for each protein across all clusters. **e** Box-and-whisker plots depicting relative frequencies of indicated immune cell clusters as a proportion of total CD45+ cells ($n = 5$ mice in each group). Data representative of two independent experiments. Comparative statistical analyses were performed using one-way ANOVA and post hoc analysis was performed using Tukey's multiple comparisons test. Data are presented as mean values ± SD. The center of the plot represents mean of the group and the whiskers represent minimum- maximum values. $p$ values for the comparisons have been indicated in the figure. $p$ values not indicated in the plot were not statistically significant. Source data are provided as a Source Data file.

background[44]. mT4-LS cells were generated and generously gifted by Dr. Michael Curran (The University of Texas MD Anderson Cancer Center, Houston, TX). B16F10 melanoma cell line was obtained from Dr. I. Fidler (The University of Texas MD Anderson Cancer Center, Houston, TX). The cells were collected in the logarithmic phase, washed twice with PBS and resuspended in 30% matrigel (Corning)/PBS just before tumor injections. 35,000 cells of mT4 or mT4-LS/50 µl and

200,000 cells of B16F10/100 µl per mouse were injected in the pancreas and intradermally, respectively.

For pancreatic orthotopic injections, the mice were anesthetized using isoflurane and injected with buprenorphine as prophylactic analgesia (3 mg/ml; i.p.). mT4/mT4-LS cells were surgically implanted in head of the pancreas using insulin syringes (29 gauge ½). Successful implantation was verified by a clear bubble formation without any

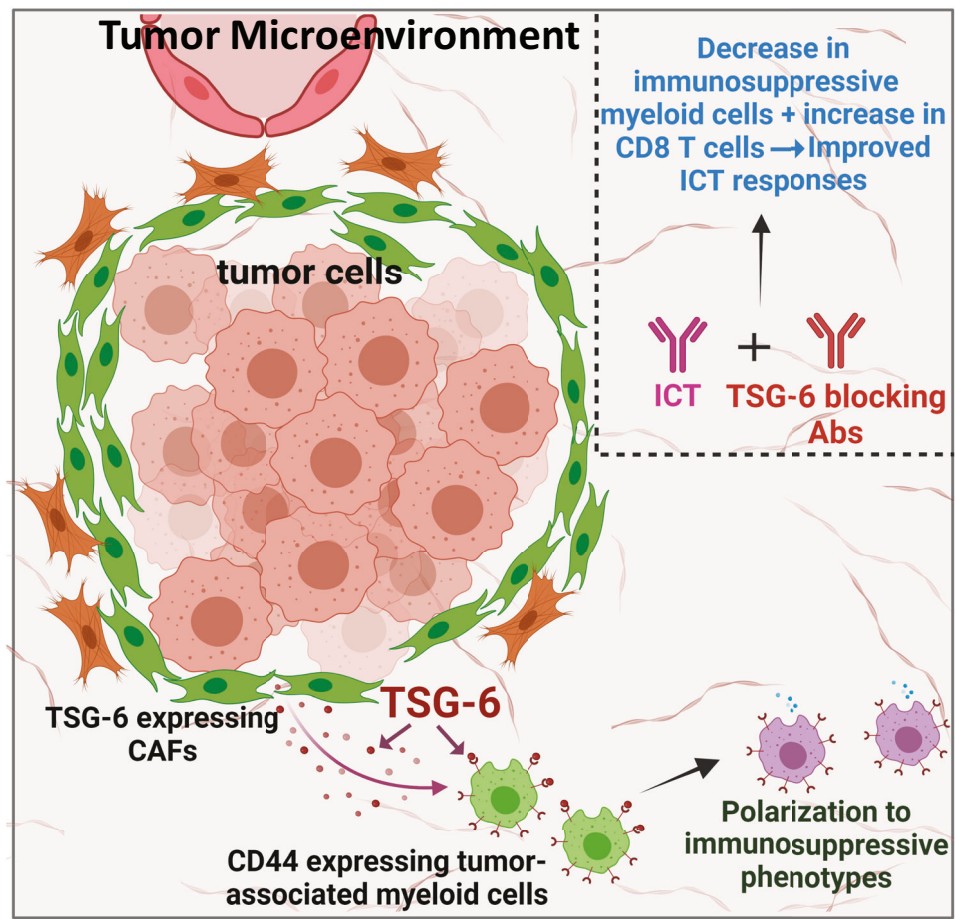

**Fig. 6 | Graphical summary describing the role of TSG-6 in ICT resistance.** Created with BioRender.com, released under a Creative Commons Attribution-NonCommercial-NoDerivs 4.0 International license.

intraperitoneal leakage. The peritoneal wall was then closed using dissolvable sutures and the skin using autoclips. The mice were kept under a heat lamp post surgery till they regained consciousness. Autoclips were removed 10 days post-surgery and mice were euthanized at the indicated time points. For survival experiments, mice were monitored daily post-surgery and euthanized humanely when they became hyperpnea, lethargic, thin or had a hunched posture to determine time of death. The maximum tumor size permitted by the Institutional Animal Care and Use Committee is 1000 mm³, and this was not exceeded for any of the experiments.

**Antibodies and treatment**
Anti-CTLA-4 (clone 9H10, cat# BP1064) and anti-PD-1 (clone RMP1-14, cat# BP0146) antibodies were purchased from BioXcell (West Lebanon, NH). Mice were injected intraperitoneally with anti-CTLA-4, anti-PD-1 and combination of anti-PD-1 plus anti-CTLA-4 on day 18 (200 μg anti-CTLA-4; 250 μg anti-PD-1/mouse), day 22 (100 μg anti-CTLA-4; 250 μg anti-PD-1/mouse) and day 26 (100 μg anti-CTLA-4; 250 μg anti-PD-1/mouse) post tumor inoculation in the mT4-LS cell line and at Day 3, 6, 9 in the parental mT4 line. The anti-TSG-6 antibody (cat# MAB2104) was purchased from R&D Systems (Minneapolis, MN). 50 μg/mouse of each antibody was injected every 4 days starting day 10 for a total of five doses.

**Tumor processing and collection**
Freshly collected murine tumors from the mice were dissociated with 0.66 mg/ml Liberase TL (Roche) and 20 mg/ml DNase I (Roche) in RPMI cell culture media and incubated for 30 min at 37 °C. Single-cell

suspensions were then made by passing digested tumors through 40 μm filters, washed in complete RPMI media, and centrifuged at 300 × g, 4 °C for 5 min. The cells were then processed as needed for the downstream analysis.

**Single-cell RNA sequencing (scRNAseq)**
Single-cell suspension of melanoma and pancreatic tumors were made using the protocol described above. Single cells were incubated with a surface staining cocktail of fluorescently conjugated antibodies, which included CD45 Pacific Blue (clone 30-F11, Biolegend, cat#103126), and live/dead discrimination viability dye Pacific Orange (Invitrogen, cat #L34968). CD45+ and CD45- cells were sorted into RPMI with 5% FBS using a FACS AriaFusion cell sorter (BD). Cell suspensions were assessed for cell concentration and viability using Life Technologies Countess 3 FL cell counter using 0.4% trypan blue exclusion staining (dead cells more permeable to staining blue). Samples passing QC fall in the concentration range for their cell target capture and have a viability of at least 70% or higher. Reagents, consumables, reaction master mixes, reaction volumes, cycling numbers, cycling conditions, and clean-up steps were completed following 10X Genomics' Next GEM 3' scRNAseq protocol. QC steps after cDNA amplification and library preparation steps were carried out by running ThermoFisher Qubit HS dsDNA Assay along with Agilent HS DNA Bioanalyzer for concentration and quality assessments, respectively. Equal amounts of each uniquely-indexed sample library was pooled together. The resultant pool was verified for concentration via qPCR using a KAPA Biosystems KAPA Library Quantification Kit. The pool was sequenced using a NovaSeq6000 sequencer depending on the total number of

samples, the target cell numbers/sample, and read depth of 50,000 read pairs/cell. The run parameters used were 28 cycles for read 1, 91 cycles for read2, 8 cycles for index1, and 0 cycles for index2 as stipulated in the protocol mentioned above. Raw sequencing data (fastq file) was demultiplexed and analyzed using 10X Genomics Cell Ranger software utilizing standard default settings and the cell ranger count command to generate HTML QC metrics and cloupe files for each sample. Further analysis can be achieved using the cLoupe files in 10X Genomics Loupe Browser (v7.0) software.

**Pre-processing of scRNAseq data.** An initial Seurat objects was created by merging the 'filtered_feature_bc_matrix' from each sample in an experiment. We kept all features present in at least 3 cells. Cells were required to have greater than 200 and less than 6000 unique features as well as a mitochondrial gene fraction less than 0.25. The RNA data was then normalized, and cell-cycle genes were regressed out by regularized negative binomial regression (SCTransform). Data was the subjected to linear dimensionality reduction using Principal component analysis (PCA). The Uniform Manifold Approximation and Projection (UMAP) dimensional reduction technique was then carried out using the first 30 principal components. The 20 nearest neighbors were estimated using the first 30 principal components and clustered. Clusters were manually annotated by evaluating the differentially expressed genes for each individual cluster against all other clusters.

For the reanalysis of the human PDAC scRNAseq dataset, the raw fastq files of the human PDAC scRNAseq dataset were downloaded from the Genome Sequence Archive (GSA: CRA001160; Project: PRJCA001063)[27]. Cellranger v3.0.2 software (10x Genomics) was used to align the sequencing reads to the human GRCh38 genome and compute the count matrix. The Seurat R package (v4.3.0) was used to perform the analysis including filtering out low-quality cells, normalizing the data, and clustering the cells. Briefly, genes presented in less 3 cells and cells with less than 200 genes or more than 6000 genes, or with more than 10% mitochondrial gene counts were excluded from downstream analysis. PCA was applied to the top 2000 highly variable genes and the first 10 components were used for constructing a KNN graph, clustering, and UMAP projection.

**Mass cytometry (CyTOF)**
Tumors were collected at Day 29 and single-cell suspensions were made as per the protocol mentioned above. The single cell suspensions underwent ficoll treatment (Histopaque-1193) with 1:1 dilution with media, centrifuged at $260 \times g$ for 12 min (no brake to maintain the density gradient) at RT, and washed with RPMI media. Cells were then counted in an automated cell counter and 3 million cells per sample were taken for CyTOF staining. Antibodies were either purchased pre-conjugated from Standard BioTools or purchased as purified unconjugated monoclonal antibodies and conjugated in-house using MaxPar X8 Polymer kits (Standard BioTools) according to the manufacturer's instructions (Supplementary Table 2). Briefly, samples were first stained for viability with 5 µM cisplatin in 5% FACS buffer for 3 min at RT, washed thrice with FACS buffer, and barcoded using the manufacturer's protocol (Standard BioTools). Around 0.75-1 million cells from each sample were then pooled into one tube and stained with cell-surface antibodies for 30 min at 4 °C. Samples were then washed, fixed (1 h), permeabilized and stained with intracellular antibodies for 30 min at 4 °C. Post staining, the samples were washed and incubated with 125 µM Iridium intercalator (Standard BioTools) in 1.6% PFA/PBS at 4 °C overnight. The cells were then washed with PBS the next day and stored until acquisition. Right before acquisition, samples were washed twice with Milli-Q water, resuspended in water containing EQ 4 element beads (Standard BioTools), and run on a Helios mass cytometer (Standard BioTools).

**Mass cytometry analysis.** Data was first demultiplexed using the Fluidigm Debarcoder software. Files were manually gated in FlowJo v10.0 by event length for singlets, live/dead discrimination and using CD45 lineage marker for immune cells. Fcs files were then loaded into R using the flowCore package as a flowset for downstream analysis. Individual mass cytometry data files (.fcs) were filtered using FlowJo to remove normalization beads, debris, doublets, and dead cells. Remaining analysis was performed in R (version 4.2.3) and the R Foundation for Statistical Computing using R packages "cytofkit"[45], "flowcore"[46], and "CATALYST" v1.22.0. Processed data was clustered according to the FlowSOM algorithm ($k = 30$) using all cell surface markers and then manually annotated[47]. Dimensionality reduction was performed using the UMAP method[48]. Differential cluster abundances and differential protein expression analyses were performed using linear models implemented through 'edgeR'[49] and 'limma'[50], respectively.

**Cluster pathway analysis and GSEA**
Wilcox tests were performed to calculate differential gene expression between the clusters or conditions of interest. Gene ranks were calculated using the resulting log2 fold change. We evaluated the HALLMARK subset of Canonical Pathways in MSigDB v7.1[51] and considered pathways with Benjamini-Hochberg adjusted $p < 0.05$ to be significant.

**Multiplex immunofluorescence assay and analysis for patient tissues**
Using the Opal multiplex immunofluorescence staining protocol 54 on a RX-BOND (Leica) autostainer, pancreatic cancer tissue, and melanoma sections were stained for CD68 (Dako-Agilent, clone PGM-1, 1:25 dilution), CD163 (Leica Biosystems, clone 10D6, 1:50 dilution), CD44 (Cell Signaling Technologies, clone E7K27, 1:800 Dilution), TSG-6 (Novusbio, clone 38637, 1:40 dilution), and PanCK (Dako-Agilent, clone AE1/3, 1:500 dilution), depending on the analysis performed (Supplementary Table 1). Subsequent visualization was performed using Akoya Opal fluorophores (480, 690, 620, 520, 570 respectively), DAPI (1:2000 dilution), and cover-slipped using Vectashield Hardset mounting medium. Slides were scanned using a Vectra/Polaris slide scanner (PerkinElmer) and images acquired at 20X magnification were spectrally unmixed using Inform software (Akoya). Cases were analyzed for cell density and infiltration analysis was performed using HALO software (Indica labs, HighPlex FL v4.04) to determine the cells/mm² area and distance between the specific cells assessed. The data were plotted using Prism v.8.0 (GraphPad Software). Statistical significance was calculated using a Wilcoxon rank-sum two-tailed/Paired t-test. $p < 0.05$ was considered statistically significant.

**Immunofluorescence imaging in murine tissues**
The staining was performed at Histowiz, Inc. Brooklyn, using the Leica Bond RX automated stainer (Leica Microsystems) platform following a GLP-ready Standard Operating Procedure. The imaging of the duplex immunofluorescent slide was performed on the Phenoimager HT (Akoya Biosystems) platform using the Opal 570, Opal 690, and Spectral DAPI Filters with a 20X objective resulting in a 0.5 micron per pixel resolution. CD44 (Cell Signaling Technologies, clone CST37279) was diluted at 1:300 in Opal Antibody Diluent/Block (Akoya Biosystems), CD68 (Abcam, clone ab125212) was diluted at 1:200 in Opal Antibody Diluent/Block (Akoya Biosystems) and spectral DAPI (Akoya Biosystems) was diluted at 1:1000 in PBS. Blocking was performed using the same Opal Antibody Diluent/Block (Akoya Biosystems) for 30 min before each primary incubation. Antigen retrieval was performed initially using citrate-based pH 6 solution (Leica Microsystems, AR9961) for 20 min at 95 °C and used as a stripping step between the two antibody incubations. Following staining the slides were airdried, and then coverslipped with Prolong Diamond (Thermo Fisher)

mounting medium. Slides are stored in a lightproof box to prevent photobleaching at a refrigerated temperature of 4 °C.

## Immunohistochemistry staining

All the staining were performed at Histowiz, Inc. Brooklyn, using the Leica Bond RX automated stainer (Leica Microsystems) using a Standard Operating Procedure and fully automated workflow. Samples were processed, embedded in paraffin, and sectioned at 4 µm. The slides were dewaxed using xylene and alcohol-based dewaxing solutions. Epitope retrieval was performed by heat-induced epitope retrieval (HIER) of the formalin-fixed, paraffin-embedded tissue using citrate-based pH 6 solution (Leica Microsystems, AR9961) for 20 min at 95 °C. The tissues were first incubated with peroxide block buffer (Leica Microsystems), followed by incubation with the rabbit polyclonal anti-alpha SMA antibody (Abcam, clone ab5694) at 1:1000 dilution for 30 min, followed by DAB rabbit secondary reagents: polymer, DAB refine and hematoxylin (Bond Polymer Refine Detection Kit, Leica Microsystems) according to the manufacturer's protocol. The slides were dried, coverslipped (TissueTek-Prisma Coverslipper), and visualized using a Leica Aperio AT2 slide scanner (Leica Microsystems) at 40X.

## Statistical analyses

For all murine experiments, data are representative of at least two to three independent experiments with 5–10 mice in each in vivo survival experiment and 3 mice for the scRNAseq experiments, the specifics of which have been indicated in the figure legends. The data were analyzed using Prism v9.0 statistical analysis software (GraphPad Software, La Jolla, CA). Student $t$ tests (two-tailed) and ANOVA were used to identify significant differences ($p < 0.05$) between treatment groups.

## Reporting summary

Further information on research design is available in the Nature Portfolio Reporting Summary linked to this article.

## Data availability

The scRNAseq datasets generated for this study have been deposited in NCBI Sequence Read Archive (SRA) repository under the BioProject accession number PRJNA1099275. The human PDAC scRNAseq publicly available data used in this study are available in the Genome Sequence Archive under the accession code CRA001160 and project PRJCA001063. Human GRCh38 genome is available under the accession code GCF_000001405.26. The remaining data are available within the Article, Supplementary Information or Source Data file. Source data are provided with this paper.

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

## Acknowledgements

P.S. is supported by the Parker Institute of Cancer Immunotherapy (PICI). S.A. is supported by the CPRIT Research Training Program (RP210028) and Ergon Foundation. The MD Anderson Cancer Center Advanced Technology Genomics Core (ATGC) Facility is supported by the grant CA016672(ATGC) and NovaSeq6000 data is supported by the NIH 1S10OD024977-01 award to the ATGC. We would also like to thank Jan Zhang, Wenbin Liu and the immunotherapy platform at MD Anderson Cancer Center for their technical assistance. Lastly, we thank Drs. Matthew Gubin and Candice Poon for their scientific input during the manuscript submission. P.S. and S.G. are members of the James P. Allison Institute.

## Author contributions

S.A. developed the project, designed and performed the experiments, analyzed data, wrote the manuscript, and acquired funding. S.M.H analyzed data, designed experiments, and wrote the manuscript. S.G. developed the project and designed the experiments. Y.C. performed bioinformatic analyses. B.G., S.M.N., L.X., A.N., D.N.T., J.L., and S.W.A. helped with the murine experiments. S.J., M.D.M. performed and analyzed IF of melanoma and pancreatic patient samples. S.B. conducted CyTOF sample processing and pre-analysis. A.M. and J.M. provided pancreatic patient tumor samples. P.S. supervised the project, provided scientific input on project development and experiments, edited the manuscript, and acquired funding support.

## Competing interests

P.S. reports consulting, advisory roles, and/or stocks/ownership for Achelois, Adaptive Biotechnologies, Affini-T, Apricity Health, BioAlta, BioNTech, Candel Therapeutics, Catalio, Dragonfly Therapeutics, Earli, Enable Medicine, Glympse, Forty-Seven Inc., Hummingbird, ImaginAb, JSL Health, Lava Therapeutics, Lytix Biopharma, Marker Therapeutics, PBM Capital, Phenomic AI, Polaris Pharma, Sporos, Time Bioventures, Trained Therapeutix, Two Bear Capital, and Venn Biosciences, and Polaris. A.M. earns royalties from Cosmos Wisdom Biotechnology, overseen by the UTMDACC Conflict of Interest Committee, and acts as a consultant for both Freenome and Tezcat Biotechnology. The remaining authors declare no competing interests.
