## [Peer Review File · Nature Communications]

TSG-6+ Cancer-Associated Fibroblasts Modulate Myeloid Cell Responses and Impair Anti-Tumor Response to Immune Checkpoint Therapy in Pancreatic CancerREVIEWER COMMENTS

Reviewer #1 (Remarks to the Author): with expertise in CAFs, pancreatic cancer

In this manuscript by Anandhan and colleagues, the authors assess mechanisms of immune checkpoint therapy resistance within the non-immune stroma. By comparing stromal features of ICT-resistant pancreatic tumors and ICT-sensitive melanoma, they identified elevated TSG-6 expression in cancer-associated fibroblasts of ICT-resistant pancreatic cancer. They propose a model wherein TSG-6 expressed by CAFs interacts with ligand CD44 on macrophages to promote immunosuppressive myeloid phenotypes and suppress ICT efficacy. The comparison of CAF features between pancreatic cancer and melanoma is innovative and the implication of TSG-6 in immune suppression in cancer is novel. In fact, TSG-6 has not been investigated in pancreatic cancer previously, further highlighting the novelty of this manuscript. The ability of anti-TSG-6 together with combo ICT to prolong survival in a pancreatic cancer mouse model compared to combo ICT alone is encouraging. However, some of the claims in the paper with respect to the immune suppression mechanism engaged by TSG-6 are not sufficiently supported by the data in the manuscript presently. Additional experiments and analyses would help to support these claims.

Specific comments:

1. TSG-6 functions in part by regulating hyaluronic acid (HA) organization, and the TSG-6 ligand CD44 is the ligand for HA. Do TSG-6-expressing CAFs physically associate with regions of greater HA content in pancreatic cancer than TSG-6-negative CAFs? Does the anti-TSG-6 treatment of tumor-bearing mice alter HA abundance or organization in vivo?
2. While the impact of recombinant TSG-6 on bone marrow-derived macrophages as shown in Figure 4e is impressive, these results should be complemented by loss-of-function experiments using CAFs genetically deficient for TSG-6 compared to control CAFs in co-culture with macrophages. Are CAFs less able to promote an immune-suppressive macrophage phenotype in the absence of TSG-6 despite expression of cytokines, chemokines, and other factors with the capacity to impact macrophage states?
3. The enrichment for CD44 expression by macrophages in pancreatic cancer compared to melanoma is difficult to appreciate based on the results in Figure 4. This result should be

validated at the protein level by co-staining for CD44 and CD68 (or another macrophage marker) on PDAC versus melanoma tissue samples.

4. The changes to macrophage states with combo ICT plus anti-TSG-6 versus combo ICT alone per Figure 5d,e appear modest. Are significant changes to established, tumor-promoting or tumor-restraining macrophage markers noted between these treatment groups such as Arg1, Mafb, Csf1r, Cd72 or others? Significant differences are hard to assess from the heatmap in Figure 5d, and these differences would be expected based on the authors' central claims.

Reviewer #2 (Remarks to the Author): with expertise in myeloid cells, pancreatic cancer

In this paper, Anandhan et al identifies that TSG-6 supports ICT resistance through regulation of immunosuppressive intratumoral myeloid cell phenotypes. The authors propose that in pancreatic cancer, but not melanoma, CAFs are a major source of TSG-6 which interacts with CD44+ myeloid cells and drives M2-like polarisation leading to suppression of anti-tumour immune response. Combination of TSG-6 neutralisation restores sensitisation to ICI, of which PDAC is classically refractory towards, primarily through a reduction in immunosuppressive myeloid cell phenotypes leading to restored CD8 T cell abundance.

Overall, the manuscript reports some interesting observations. The identification of CAF-derived TSG-6 in PDAC is novel. On the contrary, TSG-6 has previously been linked to induce an immunosuppressive phenotype in macrophages (PMID: 27911817). Some conclusions are not supported by the shown data. The way the scRNA seq data are reported raises concerns: Instead of showing global analysis of the expression data for their genes of interest (TSG6, CD44) among all the cells, the authors select only specific cell populations.

Main concerns:

Figure 1:

The authors perform scRNAseq on CD45 negative fractions from two distinct orthotopic models of PDAC and melanoma, which reveals two tumour cell populations aligning to the cancer cell lines implanted and fibroblasts. It is surprising that no other non-immune stroma cells have been detected, for example endothelial cells, or remaining non-cancerous

parenchymal cells (acinar cells, ductal cells). Thus, why did the authors not recover any other non-immune cell types in this experiment?

Related to the above, the authors emphasize the increased abundance of CAFs in the PDAC model. However, it appears that the cell counts reported in UMAP Fig 1d, e for both models are very different (overall more cells in PDAC model, hence, overall more CAFs?). The authors should report CAF abundance in proportion to recovered total cell counts.

The authors state that TSG-6 is upregulated in CAFs in PDAC. However, it remains unclear why the authors selected TSG-6 to follow up. Was TSG-6 the most differentially expressed gene? The authors should provide a volcano plot showing top DEGs comparing breast cancer vs PDAC cancer derived CAFs to support their observations.

All conclusions are based on the scRNAseq data analysis. The reported differences in fibroblast deposition should be validated by tissue staining in their models.

Figure 2:

Validation of TNFAIP6 in reanalysed human PDAC scRNAseq datasets and visualisation of TSG6+ SMA+ cells. In the tumour dataset, TNFAIP6 is enriched in the majority of fibroblasts and a minority of stellate cells. Both fibroblasts and stellate cells are known to contribute to the CAF landscape in PDAC and exist as heterogeneous populations expressing divergent levels of SMA. The authors utilise SMA as a pan-CAF marker to visualise TSG6+ CAF in tissue. Podoplanin has emerged as a pan-CAF marker in PDAC (PMID: 31197017), whereas SMA is expressed at differing levels across subtypes. The authors should provide evidence to support their choice of SMA as a CAF marker, such as including a feature plot of ACTA2 expression across fibroblast and stellate cell clusters.

Single channel images of tissue staining should be included to support the conclusion that SMA+ cells express TSG-6 as it is difficult to determine co-localisation. The labelling should be improved in Figure 2.e to identify which image belongs to PDAC & melanoma tissue. The expression of SMA appears lower in TSG-6+ cells in PDAC tissue: the authors should comment how this aligns with CAF heterogeneity.

In contrast to the conclusion in figure 1, the SMA staining in Figure 2.e would suggest similar numbers of fibroblast in PDAC and melanoma tumours. Moreover, a number of cells expressing TSG-6 exist that do not express SMA. What is the cellular origin of TSG-6 expression in these cells? The authors should use an appropriate method to confirm CAFs

are the primary source.

The authors should support their human data with further tissue staining of the mouse models to validate the scRNAseq from figure 1, or provide evidence

The title of Figure 2 “TSG-6 expression is elevated in patients with pancreatic tumors compared to melanoma patients, and it is primarily expressed by cancer associated fibroblast” is over-stated. The authors should also demonstrate the lack of expression in immune cells (using their scRNAseq data shown in figure 3).

Figure 3:

The authors state that the frequency of macrophages, monocytes, and neutrophils were nearly two-fold higher in mT4 tumours compared to B16F10 tumours, based on their transcriptional data. The authors should further support this statement with complementary methods, such as flow cytometry, tissue section analysis.

Figure 4:

CD44 is widely expressed across multiple cell types. How is CD44 expressed across other immune cells, in comparison to macrophages? The authors should show CD44 expression across all cell types, not pre-select for macrophages. Could TSG-6 be interacting with CD44 and regulating other immune cell/cancer cell functions?

Fig 4c: The analysis of PDAC patient samples is interesting. The authors should conduct a similar analysis in their mouse models to strengthen their statements.

The authors stated that TSG-6 localises with SMA+ CAFs, but SMA-TSG-6+ cells are also present. The authors state that CD44+ myeloid cells localise in closer proximity to TSG-6+, thus it should also confirm whether CD44+ myeloid cells localise in close proximity to SMA+TSG-6+ cells, and not SMA-TSG-6+ cells.

Figure 4.e, the authors generate bone marrow derived macrophages and stimulate with either LPS or TSG-6. LPS stimulation is known to potently polarise towards an immunostimulatory phenotype. The authors should include additional control groups where macrophages are not stimulated with either LPS or TSG-6 or stimulated with IL-4, to confirm whether TSG-6 is indeed inducing an immunosuppressive phenotype, or LPS is rather driving a more immunostimulatory phenotype in comparison to TSG-6.

Figure 5:

Combinatorial inhibition of TSG-6 and ICT improves ICT efficacy in mice. The authors utilise a different, less aggressive, cell line of mT4 to permit broader therapeutic window for combinatorial inhibition of TSG-6 & ICT. How is TSG-6 expressed in these tumours? Are CAFs the predominant source of TSG-6 in these tumours as suggested before? The authors should validate their findings to support the use of an alternative cell line.

Based on the CYTOF data, the authors report that TSG-6 neutralisation alone results in a decrease in immunosuppressive macrophages & Tregs, whilst increasing the proportion of CD8 T cells, but no survival benefit is gained. SMA+ CAFs are known to impair CD8 T cell infiltration, which rather accumulate at the tumour periphery, whereas immunosuppressive macrophages can regulate fibrosis. Does the reduction in immunosuppressive macrophages, by TSG-6 neutralisation, lead to reduced fibrosis and restoration of CD8 T cell infiltration? Does the combination of TSG-6 + ICI thereby permit efficient infiltration to permit anti-tumour immune function? The data would benefit with further data of the spatial localisation of reported immune cells to support the CYTOF data.

Previously, the authors state that ARG1+ macrophages dominated mT4 tumours, and TSG-6 was reported to upregulate Arg1 expression. How does TSG-6 neutralisation affect the proportion of ARG1+ macrophages in their mouse model?

Minor comments:

Figure 4E: Figure legend states “macrophages stimulated with either LPS or recombinant macrophages via qPCR”.

Gene names should be italicised throughout the figures.

Reviewer #3 (Remarks to the Author): with expertise in CAFs, cancer immunology, scRNAseq

The manuscript by Anandhan et al. identifies TSG-6 as a mediator of resistance to immune checkpoint blockade (ICB) in a pancreatic cancer model. TSG-6 was found to be primarily expressed by cancer-associated fibroblast in the tumor microenvironment (TME). TSG-6 receptor CD44 was largely restricted to tumor-associated macrophages (TAMs) and CD44 signaling activated a gene program associated with immunosuppression in vitro. Blocking

TSG-6 in combination with ICB led to a reduction of TAMs and an increase in cytotoxic CD8 T cells in a pancreatic cancer model, suggesting this might be a promising angle for combinatorial therapy.

Overall, the manuscript is of potential interest to the cancer immunotherapy field. Nevertheless, there are some key points I would like to bring to the authors' attention:

Major

- The absence of figures that appropriately quantify gene expression makes it very hard to determine if the conclusions about differences between models or clusters is fully supported by the data. Figure 1f is the only panel of figure 1 that allows a quantitative assessment (one other positive example is figure 4f, which shows violin plots - but lacks a statistical test between the models). Moreover, the UMAPs showing gene expression, such as the one in Figures 1c,e are extremely hard to read/interpret due to the color scheme chosen and the circle drawn around each point. It is hard to visually identify differences in gene expression. Last, some figure panels lack labels for expression levels, such as the color scale in figures 1e, 2d, 4a.

- It's unclear how the replicate information has been used in the single-cell analysis. The legend mentions three tumors in each group were pooled as internal control. (How was the pool deconvolved? Or were cells just pooled and replicate information was lost? How can the authors be sure that the cells in one of the groups were not biased towards one individual animal?). Given that the authors comment on abundance of populations (For example in figure 3c), these abundance estimates need to be either replicated by single-cell RNA-seq or validated by an orthogonal assay such as flow cytometry.

- No Supplementary tables are provided with marker genes for the populations identified from the scRNA-seq analysis. This does not allow the reader to interpret the clusters and markers provided in the figures and text. These three points together make it very hard to evaluate the robustness of the single-cell analysis results presented.

- I was not able to find information on the number of cells used for single-cell RNA-seq analysis in each of the experiments. Furthermore, the authors do not provide insights into how the public single-cell data from Peng et al was processed in their methods section. The UMAP in figure 2A visually looks like there are way less cells present than described by Peng et al. Or does the way the data is visualized make it look like there are fewer cells than described in the original paper? This confusion could/should be avoided by providing cell numbers and how the data from the paper by Peng et al has been processed.

- As the authors state in their introduction, multiple groups have identified subsets of CAFs with distinct expression profiles, localization and function. It would be quite essential for the reader to understand how the finding of TSG-6 expression in CAFs relates to these subsets. Is a specific subset of CAFs the main source of TSG-6? Is it the same population in human PDAC and the orthotopic PDAC model? Or does the expression pattern differ between model and patients? This requires improved analysis of the scRNA-seq data and potentially validation with staining.

- Along these lines: To validate their findings of TSG-6 enrichment in PDAC vs melanoma, the authors co-stain with SMA. SMA is not a universal fibroblast marker, but is mostly expressed by myCAF's and pericytes. Can the authors use a more general fibroblast marker, such as PDGFRA or PDPN to substantiate their findings? This might additionally help with the question of particular CAF subsets enriched in TSG-6 expression.

- Given that the efficacy data in figure 5 is generated using the mT4-LS model, it would be quite important to understand if in this model there is also a co-localization of TSG-6+ and CD68+CD44+ cells (given the co-localization data is from patients but the model could behave differently).

Other

- The statement "We observed a decrease in suppressive macrophages (VISTA+ CD206+) and regulatory T cells (Tregs), along with a concurrent increase in the abundance of CD8 T cells in the tumors treated with anti-TSG-6 and ICT antibodies" is not supported by the data

presented in figure 5e. The decrease in Tregs compared to untreated is also observed in the ICT combo and thus not a consequence of antiTSG-6 together with combo ICT. The results from the ANOVA only indicate that there is some difference between the groups, but not between which groups. No post-hoc test is shown.

- Figure panels are incorrectly cited in the main text (E.g. “To determine if these correlations translated to in vivo interactions, we performed multi-immunofluorescence on baseline pancreatic patient tissues (Figure. 4b-c)” refers to 4 c/d; “We found that the frequency of macrophages, monocytes and neutrophils was nearly two-fold higher in mT4 tumors compared to B16F10 tumors (Figure. 3d).” This refers to fire 3c).

- Capital letters for human gene symbols should be used. For example, the y-axis label of figure 1f that refers to human TCGA data provides the mouse gene symbol (Tnfaip6).

- Some figures lack a color label, such as figure 1e.

- It's unclear to me why figure 4d includes samples from PanCa and melanoma, if the proposed mechanism is primarily relevant in pancreatic cancer (according to the legend melanoma and pancreatic cancer patient data is shown).

RESPONSE TO REVIEWERS' COMMENTS

Reviewer #1 (Remarks to the Author): with expertise in CAFs, pancreatic cancer

In this manuscript by Anandhan and colleagues, the authors assess mechanisms of immune checkpoint therapy resistance within the non-immune stroma. By comparing stromal features of ICT-resistant pancreatic tumors and ICT-sensitive melanoma, they identified elevated TSG-6 expression in cancer-associated fibroblasts of ICT-resistant pancreatic cancer. They propose a model wherein TSG-6 expressed by CAFs interacts with ligand CD44 on macrophages to promote immunosuppressive myeloid phenotypes and suppress ICT efficacy. The comparison of CAF features between pancreatic cancer and melanoma is innovative and the implication of TSG-6 in immune suppression in cancer is novel. In fact, TSG-6 has not been investigated in pancreatic cancer previously, further highlighting the novelty of this manuscript. The ability of anti-TSG-6 together with combo ICT to prolong survival in a pancreatic cancer mouse model compared to combo ICT alone is encouraging. However, some of the claims in the paper with respect to the immune suppression mechanism engaged by TSG-6 are not sufficiently supported by the data in the manuscript presently. Additional experiments and analyses would help to support these claims.

Specific comments:

1. TSG-6 functions in part by regulating hyaluronic acid (HA) organization, and the TSG-6 ligand CD44 is the ligand for HA. Do TSG-6-expressing CAFs physically associate with regions of greater HA content in pancreatic cancer than TSG-6-negative CAFs? Does the anti-TSG-6 treatment of tumor-bearing mice alter HA abundance or organization in vivo?

We appreciate the reviewer's comment. Our data, both from the mouse model (Figure.1e) and patients (Figure. 2d), suggests that TSG-6 is expressed by the majority of detectable cancer-associated fibroblasts (CAFs) in pancreatic tumors; thus, making it infeasible to compare TSG+/- CAFs in the context of HA content. To investigate the potential impact of anti-TSG-6 treatment on HA abundance and organization, we conducted Alcian blue staining (specific to HA) on tumors obtained from mice treated with or without anti-TSG-6 (see Revision Figure 1), however, no significant differences were observed between these groups. Therefore, it is unlikely that TSG-6 treatment influences HA abundance or organization. Confirmatory experiments would be necessary to validate this, but such investigations extend beyond the scope of the present study.

Revision Figure 1: TSG-6 inhibition does not affect hyaluronic acid deposition within tumors.

Representative magnification images (40x) of Alcian blue stain. Pancreatic tissue samples from six mice (3 Untreated and 3 anti-TSG-6 treated) were fixed, paraffin embedded, serial sectioned, placed on glass slides, stained with Alcian blue and digitally scanned on the Aperio AT2 scanner (Leica Biosystems, Inc). The images were evaluated by a senior board-certified veterinary pathologist. Individual animal treatments were not revealed to the pathologist and the slide were evaluated in a blinded manner. Images show the blue regions (surrogate marker for HA) is in the extracellular matrix and not the cells. The staining in both groups were between mild (10-25% of section) to moderate (>25-50% of section).

2. While the impact of recombinant TSG-6 on bone marrow-derived macrophages as shown in Figure 4e is impressive, these results should be complemented by loss-of-function experiments using CAFs genetically deficient for TSG-6 compared to control CAFs in co-culture with macrophages. Are CAFs less able to promote an immune-suppressive macrophage phenotype in the absence of TSG-6 despite expression of cytokines, chemokines, and other factors with the capacity to impact macrophage states?

We express gratitude to the reviewer for their insightful feedback. In response to the reviewer's query, we evaluated TSG-6 expression in two established murine-derived pancreatic CAF cell lines, as previously described (PMID: 35216965). Regrettably, these CAF cell lines exhibited no detectable TSG-6 expression when grown in culture (see Revision Figure 2a). However, we observed a low level of TSG-6 expression in the NIH3T3 cell line (see Revision Figure 2b). Subsequently, we generated two cell lines from the parental NIH3T3 line: one with TSG-6 knockdown and another with TSG-6 overexpression. Using these cell lines, we conducted conditioned media assays to investigate the impact of TSG-6 presence on macrophage polarization. Briefly, after 48 hours of in vitro culture, we collected conditioned media from the three cell lines and exposed differentiated bone marrow-derived macrophages to each condition, assessing

changes in gene expression via qPCR. Despite our efforts, we did not observe significant differences among the groups. We speculate that the *in vitro* system may not accurately replicate the tumor microenvironment (TME) required for TSG-6 functionality, which is demonstrated in manuscript Figure 5. Currently, we are collaborating with Jackson Laboratory to develop a novel transgenic mouse model featuring specific TSG-6 deletion, intending to conduct further *in vivo* experiments elucidating the mechanism of TSG-6 action in the TME. This will be the subject of a subsequent manuscript following this study. Based on our current findings, we have opted to omit our previous *in vitro* analysis to prevent reader confusion and have moderated our assertions regarding the mechanism. In this study, our focus is to emphasize the importance of our discoveries regarding TSG-6 identification in the pancreatic TME and its involvement in ICT resistance, a novel aspect not previously explored.

Revision Figure 2: Cancer-associated fibroblasts lose ability to express TSG-6 when cultured *in vitro*. a) Western blot performed on proteins present in concentrated conditioned media (50X) from NIH3T3 (normal fibroblasts) and two clone of CAF13 cell lines. Molecular weight of TSG-6 is 30.9kDa. b) Expression level of TSG-6 in NIH3T3 cell line and the two CAF cell lines via qPCR

3. The enrichment for CD44 expression by macrophages in pancreatic cancer compared to melanoma is difficult to appreciate based on the results in Figure 4. This result should be validated at the protein level by co-staining for CD44 and CD68 (or another macrophage marker) on PDAC versus melanoma tissue samples.

We have now included the quantification of CD44 expressing CD68 macrophages at the protein level in the tissue samples (Supplementary Figure. 4b) which further validates our scRNAseq data.

4. The changes to macrophage states with combo ICT plus anti-TSG-6 versus combo ICT alone per Figure 5d,e appear modest. Are significant changes to established, tumor-promoting or tumor-restraining macrophage markers noted between these treatment groups such as *Arg1*, *Mafb*, *Csf1r*, *Cd72* or others? Significant differences are hard to assess from the heatmap in Figure 5d, and these differences would be expected based on the authors' central claims.

We thank the reviewer for their valuable input. Unfortunately, we did not have *Mafb* and *Cd72* in our CyTOF panel and therefore could not compare their expression levels. Nevertheless, we did examine the expression of *Csf1r* (CD115), *Arg1* and some other functional markers through our CyTOF, yet observed no significant differences. The primary objective of employing CyTOF was to evaluate changes comprehensively in the immune cell populations during treatment, as it would be challenging to directly attribute the effects of TSG-6 on these markers within the complex system of the TME, where multiple environmental factors influence outcomes. To address these limitations, we are currently developing a specific mouse model tailored to elucidate these unanswered questions.

Reviewer #2 (Remarks to the Author): with expertise in myeloid cells, pancreatic cancer

In this paper, Anandhan et al identifies that TSG-6 supports ICT resistance through regulation of immunosuppressive intratumoral myeloid cell phenotypes. The authors propose that in pancreatic cancer, but not melanoma, CAFs are a major source of TSG-6 which interacts with CD44+ myeloid cells and drives M2-like polarisation leading to suppression of anti-tumour immune response. Combination of TSG-6 neutralisation restores sensitisation to ICI, of which PDAC is classically refractory towards, primarily through a reduction in immunosuppressive myeloid cell phenotypes leading to restored CD8 T cell abundance.

Overall, the manuscript reports some interesting observations. The identification of CAF-derived TSG-6 in PDAC is novel. On the contrary, TSG-6 has previously been linked to induce an immunosuppressive phenotype in macrophages (PMID: 27911817). Some conclusions are not supported by the shown data. The way the scRNA seq data are reported raises concerns: Instead of showing global analysis of the expression data for their genes of interest (TSG6, CD44) among all the cells, the authors select only specific cell populations.

We extend our gratitude to the reviewer for their insightful comments and take this opportunity to elucidate the foundations of our study. Upon assessment of all the cells populations (immune and non-immune) present in the tumor microenvironment (TME), we observed that TSG-6 was predominantly expressed by the cancer-associated fibroblasts (CAFs). Therefore, we focused our analysis on the CD45 negative cells in our murine models. However, to address the reviewer's concerns, we have also included UMAP plots to underscore the absence of *Tnfaip6* expression in the immune cells in our murine scRNAseq dataset (Supplementary Figure. 3d). Additionally, we have incorporated scRNAseq analysis encompassing all cells in human tumors (Figure. 2a-d) in the manuscript which further clarifies that *TNFAIP6* is expressed predominantly by CAFs (Figure. 2d-e).

- 1) We acknowledge the reviewer's point that CD44 is expressed by multiple subsets. However, our study was designed to investigate how TSG-6 influences anti-tumor immunity through its effects on immune cells. In both the murine and human scRNAseq dataset, we observed high expression of CD44 on macrophages and therefore focused our study accordingly (Revision Figure 3; manuscript Supplementary Figure. 4a). Among the immune cells, CD44 was also expressed by neutrophils in pancreatic tumors. The investigation of TSG-6-mediated immune checkpoint therapy resistance via neutrophils and macrophages is currently underway in our lab. However, this aspect exceeds the scope of the current study. We have now included this point in our discussion section (line 220- 225).**
- 2) We concur with the reviewer's assessment that TSG-6 has been implicated in macrophage polarization. This was, in fact, a key aspect we considered in our study. Existing studies, including the one cited by the reviewer (PMID: 27911817) and by us in the manuscript, primarily explore the role of TSG-6 in inflammatory contexts rather than the cancer setting. The primary objective of our study is to unravel whether the distinct phenotypes of macrophages in pancreatic and melanoma tumors are influenced by their interactions with TSG-6, thereby contributing to immune checkpoint therapy resistance. We posit that the TME plays a crucial role in modulating TSG-6's function and therefore,**

are currently elucidating the mechanisms of TSG-6 function using a novel mouse model we are developing, which will be the subject of our subsequent manuscript.

Lastly, we hope to resolve any other concerns the reviewer might have regarding the scRNAseq data reporting with our responses below.

Revision Figure 3: scRNAseq analysis indicates high CD44 expression in murine and human macrophages. a) UMAP plot representing the immune landscape in murine B16F10 and mT4 tumors. b) Expression of *Cd44* in all immune cells present in B16F10 and mT4 tumors. Macrophages and neutrophils are highlighted in the enclosed region with dotted black line. c) UMAP plot representing the tumor immune landscape from normal pancreas and PDAC patients reanalyzed from Peng *et al.* (PMID:31273297). d) Expression of *CD44* in all cells analyzed from the human scRNAseq dataset. Macrophages are highlighted in the enclosed region with dotted black line.

Main concerns:

Figure 1:

The authors perform scRNAseq on CD45 negative fractions from two distinct orthotopic models of PDAC and melanoma, which reveals two tumour cell populations aligning to the cancer cell lines implanted and fibroblasts. It is surprising that no other non-immune stroma cells have been detected, for example endothelial cells, or remaining non-cancerous parenchymal cells (acinar cells, ductal cells). Thus, why did the authors not recover any other non-immune cell types in this experiment?

Related to the above, the authors emphasize the increased abundance of CAFs in the PDAC model. However, it appears that the cell counts reported in UMAP Fig 1d, e for both models are very different (overall more cells in PDAC model, hence, overall more CAFs?). The authors should report CAF abundance in proportion to recovered total cell counts. The authors state that

TSG-6 is upregulated in CAFs in PDAC. However, it remains unclear why the authors selected TSG-6 to follow up. Was TSG-6 the most differentially expressed gene? The authors should provide a volcano plot showing top DEGs comparing breast cancer vs PDAC cancer derived CAFs to support their observations.

All conclusions are based on the scRNAseq data analysis. The reported differences in fibroblast deposition should be validated by tissue staining in their models.

We thank the reviewer for their feedback. The reviewer has highlighted four key concerns in their comment (underlined for reference), and we have addressed each concern in separate bullet points below:

- **The reviewer raises the concern about the absence of recovery of other non-immune cells in our murine scRNAseq dataset. However, we would like to emphasize that unlike human tumors, these are orthotopic tumors derived from murine cell lines. For scRNAseq analysis, we specifically collect the tumor area, excluding surrounding tissues. This approach ensures cleaner populations of cancer cells and fibroblasts within the tumors. In contrast, patient tumors, being subject to biopsy, present challenges in collecting specific areas, leading to the detection of various non-cancerous parenchymal cells and endothelial cells, as illustrated in Figure. 2a.**
- **As per the reviewer's suggestion, we have now included the percentage of fibroblast in proportion to total recovered cells (Supplementary Figure.1d). Notably, we concur with the reviewer regarding the higher abundance of CAFs in pancreatic tumors as compared to melanoma. This is observations are further supported by our assessments in patients through TCGA data (Supplementary Figure. 1f).**
- **The reviewer raises a crucial point that requires clarification. TSG-6 is expressed by the majority of CAFs and does not exhibit differential expression between pancreatic and melanoma tumors. However, the overall abundance of CAFs is significantly higher in pancreatic tumors than in melanoma tumors, resulting in a higher TSG-6 abundance in pancreatic tumors. We apologize for any confusion caused by our previous statements and have clarified our statements in the revised manuscript (line 115-116, 232-235).**
- **Building upon the reviewer's suggestion, we have now included histology stains indicating differences in fibroblast deposition in the two tumor models at the protein level (Supplementary Figure. 1e).**

Figure 2:

Validation of TNFAIP6 in reanalysed human PDAC scRNAseq datasets and visualisation of TSG6+ SMA+ cells. In the tumour dataset, TNFAIP6 is enriched in the majority of fibroblasts and a minority of stellate cells. Both fibroblasts and stellate cells are known to contribute to the CAF landscape in PDAC and exist as heterogeneous populations expressing divergent levels of SMA. The authors utilise SMA as a pan-CAF marker to visualise TSG6+ CAF in tissue. Podoplanin has emerged as a pan-CAF marker in PDAC (PMID: 31197017), whereas SMA is expressed at differing levels across subtypes. The authors should provide evidence to support their choice of SMA as a CAF marker, such as including a feature plot of ACTA2 expression across fibroblast and stellate cell clusters.

We thank the reviewer for their input. We agree that both fibroblasts and stellate cells express SMA. However, the CAFs are the predominant source of TSG-6 (as shown by our

scRNAseq data) and therefore we logically deduced that SMA+ TSG-6+ cells as CAFs. This was also based on the spindle shaped cells we observed as the major SMA expressors in our tissues, which correlate to fibroblast like structures (Revision Figure 4, manuscript Figure. 2f). We have included this point in our discussion now (line 239-241). Further, as highlighted by the reviewer, the study conducted by Elyada *et al.* identifies Podoplanin (PDPN) as a pan-CAF marker at the RNA level (PMID: 31197107). We initially examined the expression of Podoplanin (PDPN) and SMA on patient tissues before evaluating TSG-6 expression through immunofluorescence. Our findings revealed that, at the protein level, PDPN (validated antibody) stained only a limited number of fibroblasts compared to SMA (see Revision Figure 4). Despite various studies attempting to establish specific and distinct pan-cancer CAF markers, there is currently no defined consensus, as discussed in a review by Barrett *et al.* (PMID: 33370234). In this context, SMA has been employed as a CAF marker in other prior studies as well (PMID: 24856586, 32810598).

Revision Figure 4: Podoplanin does not stain all fibroblasts present in the tumor.

A) Representative multi immunofluorescence images of pancreatic and melanoma patient tumor stained with podoplanin (PDPN) and SMA. B) quantification of SMA+ cells and C) Quantification of PDPN+ cells in three pancreatic and melanoma patients. Statistical significance was calculated using Student's t-test (two-tailed). P values have been indicated in the figure. ns; non-significant.

Elyada *et al.*'s study also identified distinct CAF subsets within PDAC tumors. However, it is crucial to highlight that, despite the authors employing specific sorting and enrichment techniques for CAFs in their scRNAseq analysis of PDAC tumors, the myofibrotic CAFs (myCAFs) expressing higher levels of SMA are more prevalent than the immunostimulatory CAFs (iCAFs) (refer to Figure 3 in the paper; PMID: 31197107). Therefore, it is unsurprising that in both our study and the human scRNAseq analysis by Peng *et al.*, where no enrichment was performed, all CAFs express SMA and exhibit myCAF-like properties based on their gene expression profile (See Revision Figure 5, manuscript Supplementary Figure. 5).

Revision Figure 5: The predominant cancer-associated fibroblasts identified in both murine and human dataset expressed myofibrotic phenotype. A) Representative UMAP plot of all CD45 negative cells present in B16F10 and mT4 tumors analyzed from scRNAseq. B) Expression of genes indicative of myofibrotic cancer-associated fibroblasts (myCAF) phenotypes. Gene set used to define the murine CAFs was obtained from Elyada *et al.* (PMID: 31197107). C) Representative UMAP plot of all cells reanalyzed from Peng *et al.* (PMID:31273297). D) Expression of genes indicative of myCAF phenotypes. Gene set used to define the human CAFs was obtained from Elyada *et al.* (PMID: 31197107).

Keeping these considerations in mind, we opted to utilize SMA as our designated CAF marker. However, in response to the reviewer's concerns, we have now incorporated the following statement into the discussion section (line 235-239): ***“Furthermore, the predominant CAF subsets in these pancreatic tumors appear to be myofibrotic (Supplementary Figure. 5), consistent with observation from previous studies (PMID: 31197107). Given that smooth muscle actin (SMA) is abundantly expressed in myofibrotic CAFs (myCAF), we specifically evaluated TSG-6 in SMA+CAF, recognizing varying expression levels in other CAF subsets (PMID: 31197107).”***

Single channel images of tissue staining should be included to support the conclusion that SMA+ cells express TSG-6 as it is difficult to determine co-localisation. The labelling should be improved in Figure 2.e to identify which image belongs to PDAC & melanoma tissue. The expression of SMA appears lower in TSG-6+ cells in PDAC tissue: the authors should comment how this aligns with CAF heterogeneity. In contrast to the conclusion in figure 1, the SMA staining in Figure 2.e would suggest similar numbers of fibroblast in PDAC and melanoma tumours. Moreover, a number of cells expressing TSG-6 exist that do not express SMA. What is the cellular origin of TSG-6 expression in these cells? The authors should use an appropriate method to confirm CAFs are the primary source.

In response to the reviewer’s suggestions, we have now incorporated the single channel images of TSG-6 and SMA from the immunofluorescence imaging and replaced them with improved images demonstrating co-localization of TSG-6 and SMA in pancreatic tumors (Figure. 2f-h). Additionally, Figure. 2e (now Figure. 2f) has also been labelled with the corresponding tumors represented by the images, and we apologize for

the oversight in the previous manuscript draft. Further, we have quantified the numbers of SMA+ cells, revealing a higher abundance of fibroblasts in PDAC compared to melanoma tumors (Figure. 2g).

We agree with the reviewer's observation that TSG-6 stains in areas where fibroblasts are absent. We hypothesize that this occurs because TSG-6 is a secretory protein, and as such, it can be present in the TME beyond fibroblasts. However, it is essential to note that our scRNAseq data implicates CAFs as the primary source of TSG-6 (Figures. 1e, 2d).

The authors should support their human data with further tissue staining of the mouse models to validate the scRNAseq from figure 1, or provide evidence

We appreciate the reviewer's feedback. Unfortunately, there is a lack of commercially available TSG-6 antibodies suitable for staining murine tissues. Despite our attempts with several antibodies, we encountered considerable non-specific staining in the tissues, making it challenging to reliably discern the expression of TSG-6 in CAFs. However, TSG-6 antibodies designed for human tissues are accessible and demonstrate TSG-6 expression in the human tissues, which we consider more important translationally.

The title of Figure 2 "TSG-6 expression is elevated in patients with pancreatic tumors compared to melanoma patients, and it is primarily expressed by cancer associated fibroblast" is overstated. The authors should also demonstrate the lack of expression in immune cells (using their scRNAseq data shown in figure 3).

We have now included all the cells (immune and non-immune) from the Peng *et al* dataset (as compared to the earlier depiction where we focused only on the non-immune cells) in Figure. 2, which clearly demonstrates that TSG-6 is predominantly expressed by the CAFs. However, we have now edited the result subheading to "TSG-6 expression is induced in cancer setting"

Figure 3:

The authors state that the frequency of macrophages, monocytes, and neutrophils were nearly two-fold higher in mT4 tumours compared to B16F10 tumours, based on their transcriptional data. The authors should further support this statement with complementary methods, such as flow cytometry, tissue section analysis.

We have now performed and included proteomic data from CyTOF analysis that further validates the differences in the immune cell populations we observe between the B16F10 and mT4 tumors through our scRNAseq analysis (Supplementary Figure. 3c).

Figure 4:

CD44 is widely expressed across multiple cell types. How is CD44 expressed across other immune cells, in comparison to macrophages? The authors should show CD44 expression

across all cell types, not pre-select for macrophages. Could TSG-6 be interacting with CD44 and regulating other immune cell/cancer cell functions?

We thank the reviewer for their input. As explained earlier, we agree with the reviewer that CD44 is widely expressed in multiple cell types, and it is therefore likely that TSG-6 could regulate other immune cells as well as cancer cell functions. In this study, we focus on how TSG-6 regulates myeloid cells and upon our assessment we observed that CD44 was abundantly expressed by macrophages (Revision Figure 3). Additionally, we also observed CD44 expression on neutrophils. However, the detailed mechanism of TSG-6-mediated neutrophil and macrophage suppression requires thorough investigation, which we have already initiated, as mentioned earlier. However, that is beyond the scope of the current study. We have now addressed these points in our discussion section (line 220-225).

Fig 4c: The analysis of PDAC patient samples is interesting. The authors should conduct a similar analysis in their mouse models to strengthen their statements.

We thank the reviewer for their suggestion. Unfortunately, there is no reliable anti-mouse anti-TSG-6 antibodies available, so we were unable to perform a similar analysis in the murine tissues.

The authors stated that TSG-6 localises with SMA+ CAFs, but SMA-TSG-6+ cells are also present. The authors state that CD44+ myeloid cells localise in closer proximity to TSG-6+, thus it should also confirm whether CD44+ myeloid cells localise in close proximity to SMA+TSG-6+ cells, and not SMA-TSG-6+ cells.

We thank the reviewer for the comment. As suggested, we have now included the infiltration analysis which indicates that the CD44+ myeloid cells localize in closer proximity to SMA+TSG-6+ cells than SMA-TSG-6+ cells (new Figure. 4e).

Figure 4.e, the authors generate bone marrow derived macrophages and stimulate with either LPS or TSG-6. LPS stimulation is known to potently polarise towards an immunostimulatory phenotype. The authors should include additional control groups where macrophages are not stimulated with either LPS or TSG-6 or stimulated with IL-4, to confirm whether TSG-6 is indeed inducing an immunosuppressive phenotype, or LPS is rather driving a more immunostimulatory phenotype in comparison to TSG-6.

We appreciate the reviewer's insight and agree that LPS is a strong stimulant and therefore could bias the data. The experiments were repeated with the indicated controls, and we failed to see a strong effect of TSG-6 alone on the bone marrow derived macrophages (BMDMs). However, the characteristics and differentiation of BMDMs can be very different from that of the macrophages observed in vivo. We speculate that the in vitro system may not accurately replicate the TME required for TSG-6 functionality, which is demonstrated in manuscript Figure. 5. To further garner mechanistic insights on TSG-6 function in the TME, we are currently in the process of generating a novel transgenic mouse in collaboration with Jackson Laboratory, with a specific deletion for TSG-6. However, this process would take us at least a year and will be the subject of a

subsequent manuscript following this study. Based on our current findings, we have opted to omit our previous in vitro analysis to prevent reader confusion and have moderated our assertions regarding the specific mechanism. In this study, our focus is to emphasize the importance of our discoveries regarding TSG-6 identification in the pancreatic TME and its involvement in ICT resistance, a novel aspect not previously explored.

Figure 5:

Combinatorial inhibition of TSG-6 and ICT improves ICT efficacy in mice. The authors utilise a different, less aggressive, cell line of mT4 to permit broader therapeutic window for combinatorial inhibition of TSG-6 & ICT. How is TSG-6 expressed in these tumours? Are CAFs the predominant source of TSG-6 in these tumours as suggested before? The authors should validate their findings to support the use of an alternative cell line.

We appreciate the reviewer's comment. It's essential to clarify that mT4-LS is not an alternative cell line but a slower-growing clone derived from the same well-characterized parental mT4 cell line, as documented in a previous study (PMID: 34341132). Since CAFs are not derived from the cell line but arise from the modification of normal fibroblasts, their characteristics remain consistent in both cases, including the continued expression of TSG-6. To further validate this, we conducted RNAscope analysis on tumor tissues from both the mT4 parental cell line and mT4-LS and saw no discernible difference in the level of TSG-6 expression in the two types of tumors (see Revision Figure 6). Therefore, to study the effects of therapeutic responses through the inhibition of TSG-6 in combination with immune checkpoint therapy, we opted to utilize the mT4-LS cell line due to its broader therapeutic window.

Revision Figure 6: TSG-6 expression is similar in both the mT4 and mT4-LS tumors. Bar plot quantification of *Tnfaip6* levels in mT4(parental) and mT4-LS tumors assessed via RNAscope. RNAscope was carried out using the ACDBio RNAscope Multiplex Fluorescent v2 standard protocol using RNAscope™ Probe- Mm-Tnfaip6 (Cat No. 507491).

Based on the CYTOF data, the authors report that TSG-6 neutralisation alone results in a decrease in immunosuppressive macrophages & Tregs, whilst increasing the proportion of CD8 T cells, but no survival benefit is gained. SMA+ CAFs are known to impair CD8 T cell infiltration, which rather accumulate at the tumour periphery, whereas immunosuppressive macrophages can regulate fibrosis. Does the reduction in immunosuppressive macrophages, by TSG-6 neutralisation, lead to reduced fibrosis and restoration of CD8 T cell infiltration? Does the combination of TSG-6 + ICI thereby permit efficient infiltration to permit anti-tumour immune function? The data would benefit with further data of the spatial localisation of reported immune cells to support the CYTOF data.

We appreciate the reviewer's comment. To clarify, our CyTOF indicates that TSG-6 alone is not sufficient to enhance CD8 T cell infiltration in the TME. However, a synergistic approach involving TSG-6 inhibition and immune checkpoint therapy (ICT) leads to increased T cell infiltration and responses within the TME. Responding to the reviewer's suggestion regarding the potential impact of TSG-6 neutralization on fibrosis reduction, we conducted Trichrome staining on untreated and anti-TSG-6 treated tumors (Revision Figure 7). Our observations revealed the presence of blue-stained fibrillar material (representing collagen fibers and fibrosis) in all samples, ranging from minimal (trace amounts <10% of section) to mild (10-25% of section) grades, with slightly higher representation in untreated mice. Consequently, while anti-TSG-6 treatment may induce a very modest reduction in fibrosis, it does not correlate with an increased CD8 T cell infiltration. Infact, the notable increase in CD8 T cell infiltration is evident in both the ICT only and anti-TSG-6 + ICT group, hence we do not posit that TSG-6 inhibition by itself plays a role in regulating CD8 T cell infiltration.

Sample Name	Trichrome
TSG-6 1	Mild
TSG-6 2	Minimal
TSG-6 3	Minimal
Unrx 1	Mild
Unrx 2	Mild
Unrx 3	Minimal

Revision Figure 7 : Modest but non-significant decrease in fibrosis was observed in tumors treated with anti-TSG-6. Representative magnification images (40x) of Trichrome stain. Pancreatic tissue samples from six mice (3 untreated and 3 anti-TSG-6 treated) were fixed, paraffin embedded, serial

sectioned, placed on glass slides, stained with Alcian blue and digitally scanned on the Aperio AT2 scanner (Leica Biosystems, Inc). The images were evaluated by a senior board-certified veterinary pathologist. Individual animal treatments were not revealed to the pathologist and the slide were evaluated in a blinded manner. The staining in both groups were between minimal (trace amounts <10% of section) to mild (10-25% of section).

Previously, the authors state that ARG1+ macrophages dominated mT4 tumours, and TSG-6 was reported to upregulate Arg1 expression. How does TSG-6 neutralisation affect the proportion of ARG1+ macrophages in their mouse model?

We thank the reviewer for the comment. While we did not find a distinct cluster of Arg1+macrophages, we saw that the macrophage subsets we identified and describe through our CyTOF analysis (VISTA+CD206+ macs and CD163+ macs) that decrease in the anti-TSG-6 + ICT groups exhibit some Arg1 expression.

Minor comments:

Figure 4E: Figure legend states “macrophages stimulated with either LPS or recombinant macrophages via qPCR”.

Gene names should be italicised throughout the figures.

We thank the reviewer for that comment. We have currently opted to omit this figure from the manuscript, as explained earlier.

Reviewer #3 (Remarks to the Author): with expertise in CAFs, cancer immunology, scRNAseq

The manuscript by Anandhan et al. identifies TSG-6 as a mediator of resistance to immune checkpoint blockade (ICB) in a pancreatic cancer model. TSG-6 was found to be primarily expressed by cancer-associated fibroblast in the tumor microenvironment (TME). TSG-6 receptor CD44 was largely restricted to tumor-associated macrophages (TAMs) and CD44 signaling activated a gene program associated with immunosuppression in vitro. Blocking TSG-6 in combination with ICB led to a reduction of TAMs and an increase in cytotoxic CD8 T cells in a pancreatic cancer model, suggesting this might be a promising angle for combinatorial therapy.

Overall, the manuscript is of potential interest to the cancer immunotherapy field. Nevertheless, there are some key points I would like to bring to the authors' attention:

Major

- The absence of figures that appropriately quantify gene expression makes it very hard to determine if the conclusions about differences between models or clusters is fully supported by the data. Figure 1f is the only panel of figure 1 that allows a quantitative assessment (one other positive example is figure 4f, which shows violin plots - but lacks a statistical test between the models). Moreover, the UMAPs showing gene expression, such as the one in Figures 1c,e are extremely hard to read/interpret due to the color scheme chosen and the circle drawn around each point. It is hard to visually identify differences in gene expression. Last, some figure panels lack labels for expression levels, such as the color scale in figures 1e, 2d, 4a.

We thank the reviewer for their comments. We have made the following edits in the scRNAseq as requested the reviewer:

- 1) Quantitative assessments have been included for other figure panels such as Figures. 2e, Supplementary Figures. 1d, 1f and 5.**
- 2) We have added statistical value to the violin plot in Figure. 4.**
- 3) The color scheme for Figure. 1c, e have been revised for easier interpretation. Further we have also removed the circles drawn around each point from all our UMAP plots, both murine and human dataset.**
- 4) Expression levels have now been indicated for Figures. 1e, 2d and 4a. Additionally, all new figures included have also been provided with expression level bars, including Supplementary Figures. 3d, 4a, and 4c.**

- It's unclear how the replicate information has been used in the single-cell analysis. The legend mentions three tumors in each group were pooled as internal control. (How was the pool deconvolved? Or were cells just pooled and replicate information was lost? How can the authors be sure that the cells in one of the groups were not biased towards one individual animal?). Given that the authors comment on abundance of populations (For example in figure 3c), these abundance estimates need to be either replicated by single-cell RNA-seq or validated by an orthogonal assay such as flow cytometry.

We appreciate the feedback from the reviewer. In our experimental approach, we combined single-cell suspensions obtained from syngeneic murine tumors before conducting the scRNAseq experiment. This decision was guided by several considerations:

- 1) Pooling tumors and treating them as a single sample/group served to mitigate technical variabilities inherent in high-sensitivity assays.
- 2) Given that these tumors were syngeneic, injected on the same day, and processed concurrently at identical time points with similar tumor sizes, we anticipated less bias in biological replicates than in technical replicates, making them suitable internal controls.
- 3) Considering the expenses associated with scRNAseq experiments, individually analyzing each tumor from every group would significantly escalate the experiment's cost, potentially by at least three-fold.

In response to the reviewer's suggestion, we conducted mass cytometry (CyTOF) analysis on tumors collected from B16F10 and mT4 tumors, which served to reinforce the differences in immune cell populations identified in our scRNAseq analysis (Revision Figure 8; manuscript Supplemental Figure. 3c). It's worth noting that the CyTOF analysis was carried out on independent biological replicates (n = 5 for B16F10 and n = 6 for mT4 tumors) and yielded consistent results with our scRNAseq findings, thereby further validating our experimental approach and analysis.

Revision Figure 8: Tumor infiltrating immune landscape in B16F10 and mT4 tumors assessed via CyTOF. Immune cell populations identified via CyTOF analysis performed on murine mT4 pancreatic tumors (n=6) and B16F10 melanoma tumors (n=5). Based on the markers, the populations were defined as CD8 T cells (CD8+ CD3e+), CD4 T cells (CD4+ CD3e+), Tregs cells (FoxP3+ CD4+ CD3e+), B cells (CD19+ cells), NK cells (NK1.1+), macrophages (CD11b+ F4/80+), monocytes (CD11b+Ly6c+F4/80-), neutrophils (CD11b+Ly6C+Ly6G+) and dendritic cells (DCs) (CD11c+MHCII+).

- No Supplementary tables are provided with marker genes for the populations identified from the scRNA-seq analysis. This does not allow the reader to interpret the clusters and markers provided in the figures and text. These three points together make it very hard to evaluate the robustness of the single-cell analysis results presented.

We thank the reviewer for their comment. The following figures indicate the expression of the markers through which we defined our populations:

- 1) Figure. 1c UMAP plot indicates the expression of the genes used to define the cells in Figure. 1b, namely, mT4 tumor cells (Krt18, 19), the B16F10 tumor cells (Pmel, Mlana) and the fibroblasts (Col1a, Dcn).**
- 2) Likewise, the violin plots in Figure. 2c depict the expression of genes utilized to delineate the cell subsets in Figure. 2a-b. This gene set aligns with the one employed by Peng *et al.* (PMID: 31273297), whose dataset we utilized, and which has been published.**
- 3) Violin plot in Figure. 3d highlights the expression of the genes that helped defined the populations in Figure. 3b.**

However, for the ease of both the reviewer and reader, we have now included a table (Supplementary Table 1) of the top 20 genes for all the populations identified from our murine scRNAseq analysis.

- I was not able to find information on the number of cells used for single-cell RNA-seq analysis in each of the experiments. Furthermore, the authors do not provide insights into how the public single-cell data from Peng *et al* was processed in their methods section The UMAP in figure 2A visually looks like there are way less cells present than described by Peng *et al.* Or does the way the data is visualized make it look like there are fewer cells than described in the original paper? This confusion could/should be avoided by providing cell numbers and how the data from the paper by Peng *et al* has been processed.

We express gratitude to the reviewer for their valuable input. To address concerns and enhance clarity, we have made the following changes:

- 1) A bar plot detailing the number of cells analyzed in each scRNAseq experiment has been included (Supplementary Figures. 1c, 2b, 3a).**
- 2) The processing method for the published dataset from Peng *et al.* has been elucidated in the methods section of the revised manuscript. Initially, we concentrated on the tumor and stromal compartment within this dataset to underscore the specific expression of TSG-6 by CAFs. However, as rightly pointed out by the reviewer, UMAP was plotted with fewer cells for representational purposes, potentially causing confusion. In response, we have revised all figures depicting the human scRNAseq dataset to showcase analysis on all cells (both immune and non-immune cells from the Peng *et al.* dataset) to mitigate any ambiguity (Figures. 2a-e, Supplementary Figures. 2 and 4c). Importantly, this adjustment does not alter any interpretations; in fact, Figure 2d-e illustrates that, among all cells in the TME (both immune and non-immune), TSG-6 is predominantly expressed by the CAFs present in tumors and is absent in the normal pancreas.**

- As the authors state in their introduction, multiple groups have identified subsets of CAFs with distinct expression profiles, localization and function. It would be quite essential for the reader to understand how the finding of TSG-6 expression in CAFs relates to these subsets. Is a specific

subset of CAFs the main source of TSG-6? Is it the same population in human PDAC and the orthotopic PDAC model? Or does the expression pattern differ between model and patients? This requires improved analysis of the scRNA-seq data and potentially validation with staining.

We thank the reviewer for their input. As mentioned in the discussion section (now line 231), we see TSG-6 expression in all CAFs in both murine (Figure. 1e) and human tumors (Figure. 2d,e). However, we would like to highlight that the CAFs we observe in these tumors, have predominantly myofibroblastic (myCAF) characteristics. To demonstrate this, we classified the CAF clusters obtained from the human scRNAseq dataset using the markers described by Elyada *et al* (PMID: 31197017) and observed that all the CAF subsets expressed genes associated with myCAF phenotypes. We observed similar observations from our murine dataset wherein we obtained only one CAF subset (see Revision Figure 5, manuscript Supplementary Figure. 5). Here, it is important to note that in the paper by Elyada *et al*, the authors had specifically enriched the tumors for fibroblasts prior to performing the scRNAseq analysis and despite that, observed myCAFs to be the predominant subset (refer to Figure 3 in the paper). Therefore, it is not surprising that in both our dataset as well as the human dataset where no such enrichment was performed, we obtained myCAFs in our analysis (see Revision Figure 5, manuscript Supplementary Figure. 5). We have addressed this point in our discussion section (line 235-239).

Revision Figure 5: The predominant cancer-associated fibroblasts identified in both murine and human dataset expressed myofibroblastic phenotype. A) Representative UMAP plot of all CD45 negative cells present in B16F10 and mT4 tumors analyzed from scRNAseq. B) Expression of genes indicative of myofibroblastic cancer-associated fibroblasts (myCAF) phenotypes. Gene set used to define the murine CAFs was obtained from Elyada *et al*. (PMID: 31197107). C) Representative UMAP plot of all cells reanalyzed from Peng *et al*. D) Expression of genes indicative of myofibroblastic cancer-associated fibroblasts (myCAF) phenotypes. Gene set used to define the human CAFs was obtained from Elyada *et al*. (PMID: 31197107).

- Along these lines: To validate their findings of TSG-6 enrichment in PDAC vs melanoma, the authors co-stain with SMA. SMA is not a universal fibroblast marker, but is mostly expressed by

myCAFs and pericytes. Can the authors use a more general fibroblast marker, such as PDGFRA or PDPN to substantiate their findings? This might additionally help with the question of particular CAF subsets enriched in TSG-6 expression.

We thank the reviewer for their input. We agree that both fibroblasts and pericytes express SMA. However, the CAFs are the predominant source of TSG-6 (as shown by our scRNAseq data) and therefore SMA+ TSG-6+ cells are essentially CAFs. This was also based on the spindle shaped cells we observed as the major SMA expressors in our tissues, which correlate to fibroblast like structures (Revision Figure 4, manuscript Figure. 2f). We have included this point in our discussion now (line 239-241).

Further, the study conducted by Elyada *et al.* identifies Podoplanin (PDPN) as a pan-CAF marker at the RNA level (PMID: 31197107). We initially examined the expression of Podoplanin (PDPN) and SMA on patient tissues before evaluating TSG-6 expression through immunofluorescence. Our findings revealed that, at the protein level, PDPN (validated antibody) stained only a limited number of fibroblasts compared to SMA (see Revision Figure 4). Despite various studies attempting to establish specific and distinct pan-cancer CAF markers, there is currently no defined consensus, as discussed in a review by Barrett *et al.* (PMID: 33370234). In this context, SMA has been employed as a CAF marker in other prior studies as well (PMID: 24856586, 32810598).

Revision Figure 4: Podoplanin does not stain all fibroblasts present in the tumor. A) Representative multi-immunofluorescence images of pancreatic and melanoma patient tumor stained with podoplanin (PDPN) and SMA. B) quantification of SMA+ cells and C) PDPN+ cells in three pancreatic and melanoma patients.

Elyada *et al.*'s study also identified distinct CAF subsets within PDAC tumors. However, it is crucial to highlight that, despite the authors employing specific sorting and enrichment techniques for CAFs in their scRNAseq analysis of PDAC tumors, the myofibroblastic CAFs (myCAFs) expressing higher levels of SMA are more prevalent than the

immunestimulatory CAFs (iCAFs) (refer to Figure 3 in the paper; PMID: 31197107). Therefore, it is unsurprising that in both our study and the human scRNAseq analysis by Peng *et al.*, where no enrichment was performed, all CAFs express SMA and exhibit myCAF-like properties based on their gene expression profile (See Revision Figure 5 above, manuscript Supplemental Figure. 5).

Keeping these considerations in mind, we opted to utilize SMA as our designated CAF marker. However, in response to the reviewer's concerns, we have now incorporated the following statement into the discussion section (line 235-239): *“Furthermore, the predominant CAF subsets in these pancreatic tumors appear to be myofibrotic (Supplementary Figure. 5), consistent with observation from previous studies (PMID: 31197107). Given that smooth muscle actin (SMA) is abundantly expressed in myofibrotic CAFs (myCAFs), we specifically evaluated TSG-6 in SMA+CAF, recognizing varying expression levels in other CAF subsets (PMID: 31197107).”*

- Given that the efficacy data in figure 5 is generated using the mT4-LS model, it would be quite important to understand if in this model there is also a co-localization of TSG-6+ and CD68+CD44+ cells (given the co-localization data is from patients but the model could behave differently).

We appreciate the reviewer's comment. Regrettably, we encountered challenges in finding suitable TSG-6 antibodies commercially available for staining murine tissues. Despite testing several antibodies, we observed considerable non-specific staining of the tissues, making it difficult to confidently determine TSG-6 expression. While we confirmed similar TSG-6 expression levels in the mT4-LS tumors compared to the parental cell line at the RNA level, technical limitations hindered our ability to perform co-localization studies of TSG-6 with CD44+ CD68+ cells with absolute certainty.

Revision Figure 6: TSG-6 expression is similar in both the mT4 and mT4-LS tumors. Bar plot quantification of *Tnfaip6* levels in mT4(parental) and mT4-LS tumors assessed via RNAscope. RNAscope was carried out using the ACDBio RNAscope Multiplex Fluorescent v2 standard protocol using RNAscope™ Probe- Mm-Tnfaip6 (Cat No. 507491).

Other

- The statement “We observed a decrease in suppressive macrophages (VISTA+ CD206+) and regulatory T cells (Tregs), along with a concurrent increase in the abundance of CD8 T cells in the tumors treated with anti-TSG-6 and ICT antibodies” is not supported by the data presented in figure 5e. The decrease in Tregs compared to untreated is also observed in the ICT combo and thus not a consequence of antiTSG-6 together with combo ICT. The results from the ANOVA only indicate that there is some difference between the groups, but not between which groups. No post-hoc test is shown.

We thank the reviewer for this insight and have now included the p value from post hoc analysis performed on these groups (Tukey’s multiple comparison test). We would like to emphasize that the post hoc analysis does not change our previous interpretations. Additionally, we agree there is a decreasing trend in Tregs in the ICT combo group but that was not significant in our analysis. However, a significant decrease in Tregs was observed in tumors treated with anti-TSG-6 only and with combination of anti-TSG-6 + combo ICT.

- Figure panels are incorrectly cited in the main text (E.g. “To determine if these correlations translated to in vivo interactions, we performed multi-immunofluorescence on baseline pancreatic patient tissues (Figure. 4b-c)” refers to 4 c/d; “We found that the frequency of macrophages, monocytes and neutrophils was nearly two-fold higher in mT4 tumors compared to B16F10 tumors (Figure. 3d).” This refers to fire 3c).

We have corrected all the figure citations now.

- Capital letters for human gene symbols should be used. For example, the y-axis label of figure 1f that refers to human TCGA data provides the mouse gene symbol (Tnfaip6).

We apologize for the oversight and have corrected the figure now.

- Some figures lack a color label, such as figure 1e.

We have incorporated the color label for 1e and for other figures as well.

- It’s unclear to me why figure 4d includes samples from PanCa and melanoma, if the proposed mechanism is primarily relevant in pancreatic cancer (according to the legend melanoma and pancreatic cancer patient data is shown).

We apologize for that oversight and typo. We have edited the figure legend in the revised manuscript to now indicate those are only pancreatic tumors.

REVIEWERS' COMMENTS

Reviewer #1 (Remarks to the Author):

The authors have meaningfully addressed my comments from the original submission.

Reviewer #2 (Remarks to the Author):

The authors have carefully addressed all my original comments. The authors provide now additional experimental controls and data where possible to support their claims or provide in their rebuttal letter a sensible explanations why additional analysis couldn't be performed.

I am satisfied with the revisions and would like to congratulate the authors to their work.

Reviewer #3 (Remarks to the Author):

The authors successfully addressed my concerns.